# Cognitive impairment and its risk factors among Myanmar elderly using the Revised Hasegawa's Dementia Scale: A cross-sectional study in Nay Pyi Taw, Myanmar

Yu Mon Saw[1,2]*, Thu Nandar Saw[3], Thet Mon Than[1,4], Moe Khaing[1,4], Pa Pa Soe[5], San Oo[6], Su Myat Cho[1], Ei Mon Win[7], Aye Myat Mon[4], Etsuko Fuchita[8], Tetsuyoshi Kariya[1,2], Shigemi Iriyama[8], Nobuyuki Hamajima[1]

1 Department of Healthcare Administration, Nagoya University Graduate School of Medicine, Nagoya, Japan, 2 Nagoya University Asian Satellite Campuses Institute, Nagoya, Japan, 3 Department of Community and Global Health, University of Tokyo, Tokyo, Japan, 4 Medical Care Division, Department of Medical Services, Ministry of Health and Sports, Nay Pyi Taw, Myanmar, 5 Department of Preventative and Social Medicine, University of Medicine, Mandalay, Myanmar, 6 Department of Neurology, Yangon General Hospital, Yangon, Myanmar, 7 Department of Public Health, Ministry of Health and Sports, Nay Pyi Taw, Myanmar, 8 Nursing Sciences Division, Department of Integrated Health Sciences, Nagoya University Graduate School of Medicine, Nagoya, Japan

* sawyumon@med.nagoya-u.ac.jp

## Abstract

### Background

Globally, elderly population with impaired cognitive function, such as dementia, has been accelerating, and Myanmar is no exception. However, cognitive function among elderly in Myanmar has rarely been assessed. This study aimed to identify the rate of cognitive impairment and its risk factors among the elderly in Myanmar.

### Methods

This cross-sectional study was conducted at rural health centers in Nay Pyi Taw Union Territory, Myanmar, from December 2018 to January 2019. In total, 757 elderly individuals aged 60 years or over (males: 246 [32.5%], females: 511 [67.5%]) were interviewed using a face-to-face method with a pre-tested questionnaire. Descriptive statistics and multivariable logistic regression analyses were performed.

### Results

The rate of impaired cognitive function among participants was 29.9% (males: 23.6%; females: 32.9%). The following participants were more likely to present cognitive impairment: those aged 70–79 years (adjusted odds ratio [AOR] = 1.8; 95% confidence interval [CI]: 1.19–2.70) and 80 years or older (AOR = 3.9; 95% CI: 2.25–6.76); those who were illiterate (AOR = 9.1; 95% CI: 3.82–21.51); and those dependent on family members (AOR = 1.6; 95% CI: 1.04–2.44). The elderly livening with their families and those who reported having good health (AOR = 0.7; 95% CI: 0.44–0.99) were less likely to have cognitive impairment.

**Data Availability Statement:** Data are available upon request as a requirement of the Institutional Review Board, University of Public Health, Yangon,

Myanmar for researchers who meet the criteria for access to confidential data. Researchers who would like to access to the data must contact Medical Care Division, Department of Medical Services, Office no. (47),Ministry of Health and Sports, Ottara Thiri Township, Ministry Zone, Nay Pyi Taw 15011, Myanmar. Tel: 95-673-411002, Fax: 95-673-411002. Email: medicalcare@mohs. gov.mm.

**Funding:** This work was supported by Grants-in-Aid for Scientific Research (No. 19K19703) from the Ministry of Education, Science, Sports, and Culture of Japan. The funder had no role in the design of the study and collection, analysis, and interpretation of data and in writing the manuscript.

**Competing interests:** The authors have declared that no competing interests exist.

## Conclusion

Using the HDS-R Myanmar version, this study reported that there out of five elderly participants had cognitive impairment, and its risk factors, altering policy makers that Myanmar needs to prepare for adequate healthcare services and social support for elderly with cognitive impairment. Future research should be performed not only to detect general cognitive impairment but also to differentiate specific cognitive domains impairments among Myanmar elderly. Longitudinal studies are needed to observe the causal and protective factors associated with cognitive impairments in Myanmar.

## Introduction

The elderly population with impaired cognitive function has been increasing worldwide and is estimated to reach 131.5 million by 2050 [1,2]. Amid the prolonged life expectancy in Asia, the aged population with cognitive impairment, such as dementia, is estimated to be 22.85 million out of 485.83 million elderly people aged 60 years or older [2,3]. Approximately 60% of the global cognitive impairment (dementia) burden is borne by developing countries; China and India contribute more than 25% of the global burden [4,5]. In India, the prevalence of a range of estimated mild cognitive impairment was between 15% and 33% [6]. In low- and middle-income countries, the proportion of people living with dementia is projected to increase from 58% in 2010 to 63% in 2030, and 71% by 2050 [7].

Numerous socio-demographic, physical, and mental conditions have been found to be associated with cognitive impairment. Older age [8,9], being female [10–12], poor marital relationship [10,13–15], low educational level in earlier life [16,17], solitary living [10,13–15], low level of physical activity [10,18–21], chronic tobacco smoking [22], alcohol consumption [23], obesity [24,25], visual impairment [26], hypertension [27], and diabetes mellitus [28,29] are important risk factors for cognitive decline. Meanwhile, high socioeconomic status [30,31]; high level of social activities [13,32]; good nutrition [33]; being free from anxiety, stress [14], or depression [15]; as well as high level of physical activity [18–21] have been observed to be protective factors against cognitive impairment.

Age and gender are unmodifiable risk factors for cognitive decline. In the normal aging process, brain volume shrinkage, especially in the prefrontal cortex, which is responsible for memory performance, starts after 40 years of age and a rapid decrease in brain volume has been observed in patients over 70 years of age [34]. Nowadays, the world's population is aging as advanced medical technological advances increase life expectancy, and age-related cognitive declination has become a major issue. Non-communicable diseases (NCDs) such as hypertension, diabetes mellitus, and obesity due to low physical activity accompany aging [24,25,29,35]. These are responsible for rapid brain aging and cerebral-vascular accidents, provoking the action of pro-inflammatory cytokines with the resultant chronic inflammation and cerebral white matter atrophy leading to cognitive impairment [24,25,34]. Cognitive impairment is also influenced by hormonal changes, and females suffer most, especially after menopause, due to decreased estrogen levels [10,12].

Learning or education, especially in childhood, enhances brain structure and development by increasing brain vascularization, synapse number, and connections, which improve cognitive function [36,37]. Higher education levels are associated with lower cognitive decline as learning creates favorable structures and neurochemical alterations in the brain [36,37]. High socioeconomic status, high physical and social activities, and less dependency are protective

factors for cognitive impairments [13,30–32]. People with high socioeconomic status generally have more social contact and activities that make them more active, less dependent, and perform higher physical activities leading to slower cognitive decline [13,30–32]. Moreover, these people can have good nutrition and can easily access the health services they need, maintaining their health in a good state that can delay cognitive declination [30–32].

On the contrary, elderly people living a solitary life and with failed marital status or unhealthy behaviors such as chronic smoking or alcohol consumption had a higher risk of developing cognitive impairments [10,13,15,22]. Elderly individuals who live alone and are widowed, divorced, or separated may have low social contact and activities that can initiate or exacerbate lower mood or depression, the high-risk factor for cognitive impairment [10,13–15]. Elderly people leading a lonely life may also harbor risky behaviors such as chronic alcohol drinking or smoking, as there is no family member to control them, which can increase cognitive impairment [10,13,15,22].

The average life expectancy in Myanmar, a Southeast Asian country with a multi-ethnic population of 51.5 million, was 64.7 years, and elderly people aged 65 years and above accounted for 5.6% of the country's population [38]. Compared with the census data in 1973 and 1983, Myanmar had a larger proportion of an aging population in the 2014 census data. Moreover, the population pyramid is changing from an expanding type to a stationary one [38]. The number of elderly individuals is estimated to increase annually [38]. Meanwhile, the prevalence of non-communicable diseases (NCDs) is increasing, accounting for the major causes of death in 2013 [39]. Health care for the elderly, including a social welfare system, is underdeveloped, and half of the primary caregivers for the elderly are daughters [40]. In addition to the significant impact on society, cognitive impairment is associated with poor prognosis and other comorbidities, such as hypertension and diabetes mellitus [27,28].

Few studies have been performed on the cognitive function assessment among elderly in Myanmar. Assessing cognitive impairment among the Myanmar elderly using the Myanmar-translated version of the Revised Hasegawa's Dementia Scale (HDS-R) is still lacking. The HDS-R Myanmar version was officially translated by Myanmar and Japanese scientists, and some modifications were made according to the local context [41]. The scale does not include questions assessing the reading and writing ability of the respondents, making it convenient to use for illiterate people and every ethnic group using different languages in Myanmar. Therefore, it can be used as a screening tool to easily detect cognitive impairment among communities even by the basic health staff, which could be quite helpful in Myanmar with limited human resources for health. Age- and sex-specific cognitive functions, as well as the influencing factors, have not yet been reported. This kind of information is extremely useful to plan programs for elderly care, treatment, and prevention of dementia strategies. Therefore, the present study aimed to identify the rate of cognitive impairment and its risk factors among the elderly in Myanmar.

## Methods

### Study area and participants

This cross-sectional study was conducted at rural health centers (RHCs) in Nay Pyi Taw Union Territory, Myanmar, from December 2018 to January 2019. Nay Pyi Taw Union Territory has two districts, namely, Ottara (North) and Dekkhina (South), each of which has eight townships. The total population of Nay Pyi Taw Union Territory was 1,160,242 in 2014. Of them, 83,747 were aged at least 60 years. Elderly individuals who lived in the study area less than six months, those who were diagnosed with cognitive impairments along with mental and physical disorders (seriously ill), those who did not understand the Myanmar language,

and those who were unable to perform simple arithmetic calculation were excluded from the study.

## Sampling procedure and data collection

The study was conducted at four townships of Nay Pyi Taw Union Territory, selected by simple random sampling using a lottery method. In total, 11 out of 26 RHCs were selected for data collection from the eight townships, depending on RHC accessibility and availability. The participants were interviewed using a face-to-face method with a pre-tested questionnaire. The questionnaire pre-test was conducted among 150 elderly individuals in Yangon Region. The questionnaire was structured into four sections: 1) socio-demographic characteristics, 2) substance use behaviors, 3) health problems, and 4) assessment of cognitive function. The participants' height and weight were measured, and body weight was recorded up to the first decimal point. Blood pressure was assessed three times, 15 minutes after interviewing, using the OMRON M6 automated blood pressure monitor. Random blood sugar testing was performed using a glucometer (Medisafe Fit Smile, MS-FR501W, TERUMO). In total, 971 elderly people were invited to participate in this survey. Of them, 811 elderly participants (males: 264 [32.6%], females: 547 [67.4%]) provided written informed consent and agreed to participate in this study. The response rate was 92.5%. After cleaning the data and removing those with missing responses to the dependent and independent variables, 757 elderly participants (males: 246 [32.5%], females: 511 [67.5%]) were considered for the final data analysis.

## Study measures

**Dependent variable.** The Revised Hasegawa's Dementia Scale (HDS-R) Myanmar version [41] was used to assess the participants' cognitive impairment. The HDS-R consists of nine questions: Q1, age (1 point); Q2, the date of interview (4 points); Q3, the place of interview (2 points); Q4, ability to repeat three familiar words (3 points); Q5, subtracting 7 from 100 for twice (2 points); Q6, backward repetition of three and four digits (2 points); Q7, recall of the three words memorized in Q4 (6 points); Q8, immediate recall of five objects in pictures shown and hidden (5 points); and Q9, listing of 10 vegetable names (5 points) [41]. The perfect score on the HDS-R is 30 points and a score of 20 points or lower is considered as an indicator of reduced cognitive function. The dependent variable was cognitive impairment, which was dichotomized into "≤20 points" (presence of cognitive impairment) and "≥21 points" (absence of cognitive impairment). The cut-off point, 20/21, was applied based on the evaluation study of HDS-R test reporting 0.90 for sensitivity and 0.82 for specificity [42].

**Independent variables.** Socio-demographic characteristics, substance use behaviors, and health problems were considered as independent variables. The current age was categorized into three groups (60–69, 70–79, and ≥80) based on the 10-year age intervals. Marital status was categorized into three groups (single, married, and separated/divorced/windowed). Education was divided into three groups according to the educational background of respondents of the elderly: middle school and above, primary school, and only read and write, and illiterate. Family type was categorized into five groups: living alone, nuclear, extended, three generations, and skip generation to learn how family structures affect the cognitive functions of the elderly.

Substance use behaviors were grouped into the following categories: non-users (never use), ex-users, occasional users, and daily users to see the effect on the cognitive functions of respondents. Self-rated health, physical activities, and vision status were divided into two categories. The nutritional status of the elderly may play an important role in impairment of cognitive function. Therefore, body mass index (BMI) was categorized as underweight, normal,

overweight, and obese. Hypertension and diabetes mellitus were grouped into two categories according to self-reported and measurement results. The measurement cutoff point of blood pressure was 140/90 mmHg (hypertension: ≥140/90 mmHg) and random blood sugar was 200 mg/dL (diabetes mellitus: ≥ 200 mg/dL).

**Statistical analysis.** Data analyses were conducted using SPSS 25.0 (IBM SPSS Inc.). Descriptive and chi-squared tests were conducted to examine the socio-demographic status, health problems, and cognitive function scores according to sex. Logistic regression was performed for the association between cognitive impairment and its risk factors. Adjusted odds ratios (AORs) were estimated to assess the strength of the associations using 95% confidence intervals (CIs) for significance testing. In all analyses, the significance level was set at $p < 0.05$ (two-tailed).

## Ethical considerations

This study was approved by the Institutional Technical and Ethical Review Board, University of Public Health, Yangon, Myanmar (letter number: UPH-IRB 2018/Research/48, issued on November 30, 2018) and the Ethical Review Committee of Nagoya University Graduate School of Medicine (approval number: 2018–0436, issued on March 3, 2019). The objectives of the study and the questionnaire contents were explained to the participants before their written informed consent was obtained. Research team members helped illiterate participants read the informed consent form. These participants were requested to mark their fingerprint if they understood the content of the informed consent form and agreed to participate in the study. If participants were incompetent to consent, consent was taken from their legal proxies or advance directives. The data were anonymous; data collection and confidentiality of all data were also carefully maintained. A number of study participants were referred to the nearest public hospitals for further investigation and treatment as needed. Furthermore, this study followed the Strengthening the Reporting of Observational Studies in Epidemiology (STROBE) guidelines (S1 Appendix).

## Results

Table 1 presents the socio-demographic characteristics of the participants. Of the total 757 participants, 246 (32.5%) were males and 511 (67.5%) females, with a mean age of 71.2 (standard deviation [SD] = 7.59) and 69.7 years (SD = 7.30), respectively. More than one-fourth of them were between 60 and 64 years old (20.7% in males and 29.7% in females), whereas only 1.2% (2% in males and 0.8% in females) were aged 90 years and older (p = 0.020). The majority (64.3%) of the participants completed primary education (p <0.001). More than half of the participants were married (51.9%) and lived with their extended families (57.8%), whereas 7.5% (4.1% in males and 9.0% in females) lived alone (p = 0.014). More than two-thirds of the participants (69.4%) reported high levels of physical activities.

Table 2 shows the substance use behaviors and health-related characteristics of the participants. Of all the respondents, 13.8% (22% of males and 10% of females) were daily smokers (p <0.001), and 26% (28.9% of males and 24.7% of females) used smokeless tobacco daily (p = 0.024). Among the males, 17.9%, 6.9%, and 1.6% were former, occasional or social, and heavy drinkers (p <0.001), respectively. Approximately one-third of the participants (30.9% of the males and 37.4% of the females) reported that their health was in very poor or poor condition, whereas 34.9% (28.8% of the males and 37.8% of the females) had two or more comorbid diseases. Regarding vision status, 7.3% of the participants (5.3% of the males and 8.2% of the females) reported having poor vision. In assessing the BMI of the participants, 29.1% (28% of the males and 29.5% of the females) were underweight, and 19.6% (17.1% of the males and

**Table 1. Socio-demographic characteristic of participants.**

| Characteristics | Total (N = 757) | | Male (n = 246) | | Female (n = 511) | | P-value[‡] |
|---|---|---|---|---|---|---|---|
| | N | % | n | % | n | % | |
| **Age** | | | | | | | 0.020 |
| 60–64 | 203 | 26.8 | 51 | 20.7 | 152 | 29.7 | |
| 65–69 | 185 | 24.4 | 63 | 25.6 | 122 | 23.9 | |
| 70–74 | 170 | 22.5 | 65 | 24.4 | 105 | 20.5 | |
| 75–79 | 95 | 12.5 | 24 | 9.8 | 71 | 13.9 | |
| 80–84 | 68 | 9.0 | 26 | 106 | 42 | 8.3 | |
| 85–89 | 27 | 3.6 | 12 | 4.9 | 15 | 2.9 | |
| ≥ 90 | 9 | 1.2 | 5 | 2.0 | 4 | 0.8 | |
| **Education** | | | | | | | <0.001 |
| Illiterate | 165 | 21.8 | 11 | 4.5 | 154 | 30.1 | |
| Primary school | 487 | 64.3 | 173 | 70.3 | 314 | 61.4 | |
| Middle school | 72 | 9.5 | 41 | 16.7 | 31 | 6.1 | |
| High school | 22 | 2.9 | 16 | 6.5 | 6 | 1.2 | |
| University and above | 11 | 1.5 | 5 | 20.0 | 6 | 1.2 | |
| **Marital status** | | | | | | | <0.001 |
| Single | 49 | 6.5 | 13 | 5.3 | 36 | 7.0 | |
| Married | 393 | 51.9 | 183 | 74.4 | 210 | 41.1 | |
| Others | 315 | 41.6 | 50 | 20.3 | 265 | 51.9 | |
| **Family type** | | | | | | | 0.014 |
| Living alone | 56 | 7.5 | 10 | 4.1 | 46 | 9.0 | |
| Nuclear | 108 | 14.3 | 47 | 19.1 | 61 | 11.9 | |
| Extended | 438 | 57.8 | 144 | 58.5 | 294 | 57.5 | |
| Three generation | 113 | 14.9 | 32 | 13.0 | 81 | 15.9 | |
| Skip generation | 42 | 5.5 | 13 | 5.3 | 29 | 5.7 | |
| **Low physical activities** | | | | | | | 0.034 |
| No | 232 | 69.4 | 158 | 64.2 | 367 | 71.8 | |
| Yes | 525 | 30.6 | 88 | 35.8 | 144 | 28.2 | |
| **Place of interview** | | | | | | | 0.412 |
| Ngan Sat RHC | 60 | 7.9 | 26 | 10.6 | 34 | 6.7 | |
| Tha Pyay Pin RHC | 97 | 12.8 | 26 | 10.6 | 71 | 13.9 | |
| Zee Kone RHC | 81 | 10.7 | 29 | 11.8 | 52 | 10.2 | |
| Nat Tha Ye RHC | 92 | 12.2 | 30 | 12.2 | 62 | 12.1 | |
| Taung Po Thar RHC | 54 | 7.2 | 18 | 7.3 | 36 | 7.0 | |
| Ma Dot Pin RHC | 45 | 5.9 | 12 | 4.9 | 33 | 6.5 | |
| Baw Di Gone RHC | 60 | 7.9 | 19 | 7.7 | 41 | 8.0 | |
| Tha Wut Hti RHC | 70 | 9.2 | 27 | 11.0 | 43 | 8.4 | |
| Nyaung Lont RHC | 87 | 11.5 | 31 | 12.6 | 56 | 11.0 | |
| Pyi San Aung RHC | 49 | 6.5 | 11 | 4.5 | 38 | 7.4 | |
| Si Pin Thar Yar RHC | 62 | 8.2 | 17 | 6.8 | 45 | 8.8 | |

[‡]A chi-square test for the different between males and females. RHC: rural health center.

20.7% of the females) were obese. Among the males, 67.9% had hypertension, and 19.9% had diabetes mellitus. In females, 57.3% had hypertension, and 21.3% had diabetes mellitus.

Table 3 presents the HDS-R individual item scores for dementia by age group. The total mean score was 22.4. The highest mean score was found in the age group of 60–64 years

**Table 2. Substance use behaviors and health-related characteristic of participants.**

| Characteristics | Total (N = 757) | | Male (n = 246) | | Female (n = 511) | | P-value[‡] |
|---|---|---|---|---|---|---|---|
| | N | % | n | % | n | % | |
| **Smoking** | | | | | | | <0.001 |
| Never smoke | 516 | 68.2 | 125 | 50.8 | 391 | 76.5 | |
| Ex-smoker | 102 | 13.5 | 50 | 20.3 | 52 | 10.2 | |
| Occasional smoker | 34 | 4.5 | 17 | 6.9 | 17 | 3.3 | |
| Daily smoker | 105 | 13.8 | 54 | 22.0 | 51 | 10.0 | |
| **Smokeless tobacco use** | | | | | | | 0.024 |
| Never use | 457 | 60.4 | 139 | 56.5 | 318 | 62.2 | |
| Ex-user | 27 | 3.6 | 15 | 6.1 | 12 | 2.3 | |
| Occasional user | 76 | 10.0 | 21 | 8.5 | 55 | 10.8 | |
| Daily user | 197 | 26.0 | 71 | 28.9 | 126 | 24.7 | |
| **Alcohol drinking** | | | | | | | <0.001 |
| Never Drink | 684 | 90.4 | 181 | 73.6 | 503 | 98.4 | |
| Ex-drinker | 52 | 6.9 | 44 | 17.9 | 8 | 1.6 | |
| Occasional/social drinker | 17 | 2.2 | 17 | 6.9 | 0 | 0 | |
| Heavy drinker | 4 | 0.5 | 4 | 1.6 | 0 | 0 | |
| **Self-rated health** | | | | | | | 0.105 |
| Very poor/poor | 267 | 35.3 | 76 | 30.9 | 191 | 37.4 | |
| Fair | 237 | 31.3 | 76 | 30.9 | 161 | 31.5 | |
| Good/very good | 253 | 33.4 | 94 | 38.2 | 159 | 31.1 | |
| **No. of comorbidity** | | | | | | | 0.055 |
| No. disease | 151 | 19.9 | 54 | 22.0 | 97 | 19.0 | |
| At least one disease | 342 | 45.2 | 121 | 49.2 | 221 | 43.2 | |
| Two or more diseases | 264 | 34.9 | 71 | 28.8 | 193 | 37.8 | |
| **Vision status** | | | | | | | 0.254 |
| Good | 371 | 49.0 | 128 | 52.0 | 243 | 47.6 | |
| Fair | 331 | 43.7 | 105 | 42.7 | 226 | 44.2 | |
| Poor | 55 | 7.3 | 13 | 5.3 | 42 | 8.2 | |
| **BMI** | | | | | | | 0.126 |
| Under weight (11.9–18.4) | 220 | 29.1 | 69 | 28.0 | 151 | 29.5 | |
| Normal (18.5–22.9) | 283 | 37.3 | 106 | 43.1 | 177 | 34.6 | |
| Overweight (23.0–24.9) | 106 | 14.0 | 29 | 11.8 | 77 | 15.2 | |
| Obese ($\geq$25) | 148 | 19.6 | 42 | 17.1 | 106 | 20.7 | |
| **Hypertension** | | | | | | | 0.005 |
| No | 297 | 39.2 | 79 | 32.1 | 218 | 42.7 | |
| Yes | 460 | 60.8 | 167 | 67.9 | 293 | 57.3 | |
| **Diabetes mellitus** | | | | | | | 0.654 |
| No | 599 | 79.1 | 197 | 80.1 | 402 | 78.7 | |
| Yes | 158 | 20.9 | 49 | 19.9 | 109 | 21.3 | |

[‡]A chi-square test for the different between males and females. BMI: body mass index

(mean = 24.0) followed by 65–69 years (mean = 23.3). The lowest mean score was found in the $\geq$ 90 years age group (mean = 17.6). The mean scores for temporal orientation, spatial orientation, registration (words), attention/calculation, digit span backward, recall (words), registration (objects), and word fluency were highest in the 60–64 years age group followed by the 65–69 years group.

Table 3. Revised Hasegawa's Dementia individual item score by age group.

| Items | Total (N = 757) | 60–64 (n = 203) | 65–69 (n = 185) | 70–74 (n = 170) | 75–79 (n = 95) | 80–84 (n = 68) | 85–89 (n = 27) | ≥ 90 (n = 9) |
|---|---|---|---|---|---|---|---|---|
| | Mean (SD) | Mean (SD) | Mean (SD) | Mean (SD) | Mean (SD) | Mean (SD) | Mean (SD) | Mean (SD) |
| **Total score** | 22.4(4.4) | 24.0 (3.5) | 23.3 (3.9) | 22.3 (4.4) | 22.3 (4.1) | 20.4 (5.0) | 19.0 (5.0) | 17.6 (4.7) |
| **Age** | 1.0 (0.2) | 0.99 (1.0) | 1.0 (0.2) | 0.2 (0.2) | 1.0 (0.2) | 1.0 (0.1) | 1.0 (0.1) | 1.0 (0.1) |
| **Temporal orientation** | 2.5 (1.3) | 2.83(1.2) | 2.7 (1.3) | 2.5 (1.4) | 2.0 (1.4) | 2.3 (1.4) | 1.7 (1.3) | 2.3 (1.6) |
| **Spatial orientation** | 1.7 (0.41) | 1.9 (0.3) | 1.9 (0.2) | 1.9 (0.4) | 1.7 (0.5) | 1.8 (0.6) | 1.7 (0.6) | 1.9 (0.3) |
| **Registration (words)** | 2.3 (1.0) | 2.3 (1.0) | 2.3 (1.0) | 2.3 (1.0) | 2.2 (1.1) | 2.2 (1.1) | 1.9 (1.2) | 1.9 (1.3) |
| **Attention/Calculation** | 0.8 (0.9) | 0.9 (0.9) | 0.9 (0.9) | 0.8 (0.9) | 0.4 (0.7) | 0.6 (0.8) | 0.6 (0.8) | 0.7 (0.9) |
| **Digit span backward** | 0.5 (0.7) | 0.6 (0.8) | 0.5 (0.8) | 0.4 (0.7) | 0.3 (0.5) | 0.3 (0.5) | 0.2 (0.5) | 0.2 (0.7) |
| **Recall (words)** | 4.4 (1.8) | 4.9 (1.5) | 4.6 (1.6) | 4.3 (1.9) | 4.2 (1.9) | 4.0 (2.1) | 3.3 (2.0) | 2.2 (1.7) |
| **Registration (objects)** | 4.5 (0.8) | 4.6 (0.6) | 4.6 (0.8) | 4.6 (0.8) | 4.3 (0.9) | 4.2 (1.1) | 4.2 (0.8) | 3.4 (1.4) |
| **Word fluency** | 4.6 (0.9) | 4.8 (0.6) | 4.7 (0.8) | 4.6 (0.9) | 4.3 (1.1) | 4.1 (1.3) | 4.4 (1.3) | 4.0 (1.0) |

Table 4 lists the multivariate logistic regression analysis results of factors associated with cognitive impairment in both male and female. In bivariate analysis, the following characteristics were positively associated with cognitive impairment: being older than 70 years, female (unadjusted odds ratio [UOR] = 1.6; 95% CI: 1.12–2.25), separated or divorced or widowed (UOR = 2.7; 95% CI: 1.30–5.61), illiterate (UOR = 14.2; 95% CI: 6.45–31.08) or having the ability to only read and write (UOR = 4.4; 95%: CI 2.07–9.24), dependent (UOR = 2.5; 95% CI: 1.78–3.63), and a former substance user (UOR = 1.4; 95% CI: 1.01–1.95). Meanwhile, the following characteristics were negatively associated with cognitive impairment: belonging to any type of nuclear (UOR = 0.3; 95% CI: 0.13–0.52) or extended (UOR = 0.4; 95% CI: 0.25–0.76) or three-generation (UOR = 0.4; 95% CI: 0.20–0.77) or skip-generation family (UOR = 0.4; 95% CI: 0.17–0.94); and being either underweight (UOR = 0.7; 95% CI: 0.45–0.95) or obese (UOR = 0.4; 95% CI: 0.26–0.70).

According to adjusted analysis, participants who were in the age group of 70–79 (AOR = 1.8; 95% CI: 1.19–2.70) and 80 years or older (AOR = 3.9; 95% CI: 2.25–6.76); who were illiterate (AOR = 9.1; 95% CI: 3.82–21.51) or completed only primary level education (AOR = 3.4; 95% CI: 1.56–7.52), and dependent on family members (AOR = 1.6; 95% CI: 1.04–2.44) were more likely to have cognitive impairment. Meanwhile, participants who belonged to a nuclear (AOR = 0.4; 95% CI: 0.18–0.97), extended (AOR = 0.5; 95% CI: 0.27–0.97), or three-generation family (AOR = 0.4; 95% CI: 0.21–0.94), and who reported being in good health (AOR = 0.7; 95% CI: 0.44–0.99) were less likely to have cognitive impairment (Table 4).

## Discussion

This study is the first to examine the rate of impaired cognitive function using the HDS-R and related comorbidities among the elderly in Myanmar. The rate of impaired cognitive function among participants was 29.9% (males: 23.6%, females: 32.9%). The results revealed that female were significantly more likely to develop cognitive impairment than male participants. Participants, 70 years or older, who had a low education level (i.e., who had primary level education or who could only read and write), and who were dependent on their family members were found to be at higher odds of developing cognitive impairment. Meanwhile, participants who lived with their families and who reported being in good health were less likely to develop cognitive impairment.

In this study, the participants older than 70 years had a higher odds of developing cognitive impairment compared with the 60–69 years old age group. This finding is consistent with

**Table 4.  Multivariable logistic regression analysis of factors associated with cognitive impairment among Myanmar elderly (N = 757).**

| Characteristics | OR | 95% CI | AOR† | 95% CI |
|---|---|---|---|---|
| **Age** | | | | |
| 60–69 | | | | |
| 70–79 | 2.3 | (1.63–3.33)*** | 1.8 | (1.19–2.70)** |
| ≥80 | 4.9 | (3.07–7.72)*** | 3.9 | (2.25–6.76)*** |
| **Gender** | | | | |
| Male | | | | |
| Female | 1.6 | (1.12–2.25)** | 1.1 | (0.69–1.73) |
| **Marital status** | | | | |
| Single | | | | |
| Married | 1.1 | (0.53–2.31) | 1.1 | (0.48–2.46) |
| Separated/Divorced/Windowed | 2.7 | (1.30–5.61)** | | |
| **Education** | | | | |
| Middle school and above | | | | |
| Only read and write /Primary school | 4.4 | (2.07–9.24)*** | 3.4 | (1.56–7.52)** |
| Illiterate | 14.2 | (6.45–31.08)*** | 9.1 | (3.82–21.51)*** |
| **Dependent** | | | | |
| No | | | | |
| Yes | 2.5 | (1.78–3.63)*** | 1.6 | (1.04–2.44)* |
| **Family type** | | | | |
| Living alone | | | | |
| Nuclear | 0.3 | (0.13–0.52)*** | 0.4 | (0.18–0.97)* |
| Extended | 0.4 | (0.25–0.76)** | 0.5 | (0.27–0.97)* |
| Three generation | 0.4 | (0.20–0.77)** | 0.4 | (0.21–0.94)* |
| Skip generation | 0.4 | (0.17–0.94)* | 0.6 | (0.22–1.45) |
| **Alcohol, smoking and smokeless tobacco use** | | | | |
| Non-user (Never use) | | | | |
| Ex-user | 1.4 | (1.01–1.95)* | 1.3 | (0.89–1.88) |
| Occasional user | 1.6 | (0.88–2.89) | 1.6 | (0.81–3.30) |
| Daily user | 1.1 | (0.27–4.02) | 1.2 | (0.25–5.47) |
| **Self-rated health** | | | | |
| Very poor/poor/fair | | | | |
| Good/very good | 0.7 | (0.53–1.03) | 0.7 | (0.44–0.99)* |
| **No. of comorbidity** | | | | |
| No. diseases | | | | |
| At least one disease | 1.0 | (0.65–1.49) | 0.8 | (0.49–1.34) |
| Two or more diseases | 0.9 | (0.61–1.45) | 0.9 | (0.50–1.57) |
| **Low physical activities** | | | | |
| No | | | | |
| Yes | 0.9 | (0.66–1.29) | 1.3 | (0.88–1.90) |
| **Vision status** | | | | |
| Good | | | | |
| Fair/poor | 1.2 | (0.85–1.58) | 0.8 | (0.58–1.21) |
| **BMI $** | | | | |
| Underweight | | | | |
| Normal | 0.7 | (0.45–0.95)* | 0.9 | (0.60–1.41) |
| Overweigh | 0.8 | (0.51–1.35) | 1.4 | (0.77–2.24) |
| Obese | 0.4 | (0.26–0.70)* | 0.8 | (0.44–1.40) |

*(Continued)*

**Table 4.** (Continued)

| Characteristics | OR | 95% CI | AOR† | 95% CI |
|---|---|---|---|---|
| **Hypertension** | | | | |
| No | | | | |
| Yes | 0.7 | (0.54–1.02) | 0.9 | (0.58–1.29) |
| **Diabetes mellitus** | | | | |
| No | | | | |
| Yes | 0.8 | (0.55–1.21) | 0.9 | (0.57–1.41) |

* p<0.05

**p<0.01, p<0.001

§BMI: Underweight (11.9–18.4), Normal (18.5–22.9), Overweigh (23.0–24.9), and Obese (≥25).

†Adjusted for age, gender, marital status, education, dependent, family type, alcohol, smoking and smokeless tobacco use, self-rated health, no. of comorbidity, low physical activities, vision status, BMI, hypertension, and diabetes mellitus."

those of other studies [8,9,27,28]. The rate of cognitive impairment is the highest in the age group of 85 years and older, ranging from 16.7% in China [8] to 43% in Germany [9]. Studies have also estimated that elderly aged 75 years or older account for 80% of patients with dementia [27,28]. Aging is associated with several changes in brain structure and function. Brain volume shrinkage started around or after 40 years, and the shrinkage rate increased especially for those over 70, even in the normal aging process [34]. The most affected area is the prefrontal cortex, which is responsible for memory performance. Reduction in cortical volume associated with increased white matter lesions in the elderly leads to executive function declination and cognitive impairment [34]. Moreover, aging is usually associated with NCDs such as hypertension and type 2 diabetes mellitus [29,35]. Hypertension and diabetes mellitus account for small or large vascular changes leading to cerebral-vascular accidents, strokes, cerebral hemorrhage or micro-cerebral infarcts. As a result, cognitive function impairment is inevitable due to brain volume reduction [29,34,35]. Older age is the greatest unmodifiable risk factor for cognitive impairment [16], and should be taken into consideration in cognitive impairment prevention, as life expectancy is increasing worldwide. Aging can also combine with other comorbidities and can exacerbate the situation [16]. As such, health policymakers and stakeholders in Myanmar should initiate community preventive measures or strategies for cognitive impairment as early as possible to mitigate its adverse consequences.

Participants who had a low educational level (i.e., those who could only read and write or only had primary-level education) were found to be associated with a higher odds of developing cognitive impairment. Numerous studies have pointed out that education has a strong correlation with cognitive impairment; a low educational level in early life is associated with poor cognitive function in both cohort and cross-sectional studies worldwide [16–17]. Several mechanisms have been suggested for this correlation. One is that learning or education in early life may affect brain development and structure by enhancing synapse number and connections, as well as by increasing brain blood flow or vascularization, which could enhance cognitive function [36]. Another mechanism is that continuous mental stimulation through learning or education may increase favorable structural or neurochemical alterations in the brain, which in turn improve cognitive function [37]. However, a number of cohort studies did not find associations between low education and cognitive decline [43,44].

Participants who were dependent were more likely to have cognitive impairment. One explanation is that they may have no formal job, leading to a low level of daily physical activities, which is considered to worsen the cognitive performance of the individual [10,18–21]. In

addition, participants who were dependent were more likely to be cared for by their daughters; half of the caregivers in Myanmar were elderly' daughters [40]. These participants were considered to have a low level of physical activities; in Myanmar, daughters' care of their parents are commonly of an over-protective nature. The elderly are considered to be vulnerable to slips and falls owing to aging, and the daughters are more likely to take care of all the daily chores, which may lead to the elderly being physically less active than their independent counterparts, which in turn results in their cognitive decline [10,18–21]. Therefore, educational programs in Myanmar for daily physical exercises or activities targeting not only the elderly but also the caregivers will be beneficial.

Participants who lived with their families, either in a nuclear, extended, or three-generation type, were found to have a lower odds of developing cognitive impairment. Studies have also found that elderly who live alone have a much higher risk for developing cognitive impairment compared with elderly living with their families [10,13–15]. The elderly living alone are considered to have less social contact, which is associated with increased cognitive decline owing to lower mood and depression, compared with elderly living with family [14,15]. Notably, such a protective association was not observed in the participants who belonged to the skip generation family type in this study. One of the possible reasons is that the elderly experienced much worry and anxiety taking care of their grandchildren when their children were away, especially migrant workers, thereby leading to cognitive decline attributable to stress [14]. As such, a strong social support system that meets the needs of the elderly should be formulated, assessed, and implemented in Myanmar.

Participants who rated their health status as in good or very good condition were also less likely to develop cognitive impairment, compared with those who rated their health status as poor. One explanation may be that these elderly individuals may have a high socio-economic status and easy access to the health services they need, both of which are strong protective factors against cognitive impairment [30,31]. High socio-economic status also means having a high nutritious food intake [31], as well as a high level of social activities or communication in their lives, which have been proven to improve cognitive function [13,32]. Another alternative explanation according to the local context is that the elderly never may have done the necessary health check-up, as they appeared to be healthy, and therefore, they rated their health status as in good condition.

Among the participants, female participants and those either separated, divorced, or widowed had a higher odds of developing cognitive impairment in the unadjusted analysis. Studies have noted that sex plays a role in cognitive functioning [10,11], and females are found to have higher cognitive impairment compared with males [10], or decreased cognitive performance in patients with Alzheimer's disease [12]. A possible reason may be the hormonal difference and influence between the sexes; the decreased estrogen level in later life has a negative effect on cognitive function in females [12]. Females tend to have a longer life expectancy than males in Myanmar, but this does not necessarily apply for healthy life expectancy [45]. A longer life expectancy with high co-morbidities (i.e., aging with unproductive life) will result in a greater burden for the patient and the caregivers. Therefore, preventive strategies that account for sex differences are highly recommended to reduce the burden of cognitive impairment among Myanmar elderly. In addition, among all participants, those who were either separated, divorced, or widowed had a higher odds of developing cognitive impairment as they may have less social contact or fewer activities within the community, leading to social isolation, which is the greatest risk factor for cognitive impairment [10,13–15].

Participants whose BMI was in the normal or obese category were observed to have a lower odds of developing cognitive impairment in this study. Many studies agree that maintaining normal body weight throughout the life span is considered to be a protective factor for

cognitive impairment, which is consistent with this study [24,25]. Increased BMI associated with central obesity in the middle years of life has been observed as one of the risk factors for cognitive decline in older age by recent systematic reviews and meta-analyses [24,25]. Obesity in the middle years of life is usually related to hypertension, stroke, diabetes mellitus, and dyslipidemia [24,25,29,35]. Moreover, middle-age obesity is more likely to be associated with rapid brain aging and cerebral white matter atrophy due to the action of pro-inflammatory cytokines causing chronic inflammation and metabolic diseases [24,25,34]. Pathophysiological changes in obesity are considered to be related mainly to adiposity distribution, which cannot be measured directly by BMI as it fails to differentiate muscles from adipose tissues [24,25]. This study used BMI to categorize underweight, normal, overweight, and obese among the participants. This is one of the possible reasons why obese participants in this study had a lower odds of cognitive impairment. Another reason for the lower odds of cognitive impairment among obese participants in this study is that their obesity may start in their late-life as obesity in the later years of life (over 76 years of age) is found to be associated with slower cognitive decline in some studies [24,25].

In this study, 13.8% of the participants were daily smokers and 0.5% were heavy drinkers. Chronic smokers were more likely to be alcohol drinkers, and chronic tobacco smoking is associated with cognitive decline and the development of neurocognitive diseases in later life [22]. However, this association was not found in this study. This may be due to the limited number of study participants and the study conducted in the same demographic area in Myanmar.

This study has several strengths and limitations. This is the first study to investigate the prevalence of impaired cognitive function using the Myanmar-translated version of the HDS-R and its related comorbidities among Myanmar elderly. It was observed that 23.6% of males and 32.9% of females in this study had cognitive impairment detected by the HDS-R. The HDS-R has its own advantages. As it does not include questions assessing the reading and writing ability of respondents, it is convenient to use in illiterate people or in minor ethnic groups who cannot read and write the Myanmar language. Cognitive impairment could be checked easily by basic health staff using the HDS-R. However, cognitive impairment among the participants in this study was not confirmed by a psychiatrist. Therefore, cognitive impairment in specific domains such as executive function, spatial working memory, processing speed, attention, and verbal memory or verbal fluency could not be ruled out regardless of HDS-R allowance to assess cognitive impairment. Furthermore, the fairly wide confidence intervals were observed for some predictors despite the large sample size.

The study was conducted in one region in the central part of Myanmar, which limited the generalizability of the results, given that different socio-demographic features are represented across Myanmar. The causal relationship between cognitive impairment and different risk factors could not be explored clearly owing to the cross-sectional design of the study.

## Conclusion

Three out of five elderly participants reported having cognitive impairment and female participants were significantly more likely to develop cognitive impairment. Being over 70 years old, a low educational level, dependency, solitary living, and poor self-rated health were associated with a higher odds of developing cognitive impairment. Meanwhile, living with one's family and having good self-rated health were protective factors. Based on scientific evidence, policymakers need to consider implementing community preventive measures or strategies regarding cognitive impairment and gender differences as early as possible to mitigate its adverse consequences among Myanmar elderly. Screening for cognitive impairment using the

Myanmar language version of the HDS-R should be confirmed by the clinical diagnosis in the further studies so that even the basic health staff can screen for cognitive impairment among the general population at the most basic lever. Therefore, it could be helpful in limited health workforce settings. Future research should be performed not only to detect general cognitive impairment but also to differentiate specific cognitive domains impairments among the Myanmar elderly. Longitudinal studies are needed to observe the causal and protective factors associated with cognitive impairments and associated comorbidities in Myanmar.

## Supporting information

**S1 Appendix. STROBE (Strengthening the reporting of observational studies in Epidemiology) checklist.**
(PDF)

## Acknowledgments

The authors would like to express our sincere appreciation to all the health staff from rural health center, Nay Pyi Taw Union Territory and staff from dept. of Medical Services, Ministry of Health and Sports, Nay Pyi Taw, Myanmar for their kind support and active cooperation in this study.

## Author Contributions

**Conceptualization:** Yu Mon Saw, Thu Nandar Saw, Thet Mon Than, Moe Khaing, Pa Pa Soe, San Oo, Aye Myat Mon, Tetsuyoshi Kariya, Nobuyuki Hamajima.

**Data curation:** Yu Mon Saw, Thet Mon Than.

**Formal analysis:** Yu Mon Saw, Thu Nandar Saw, Thet Mon Than, Etsuko Fuchita, Shigemi Iriyama, Nobuyuki Hamajima.

**Funding acquisition:** Yu Mon Saw.

**Investigation:** Yu Mon Saw, Thu Nandar Saw, Thet Mon Than, Moe Khaing, Pa Pa Soe, San Oo, Su Myat Cho, Ei Mon Win, Aye Myat Mon, Tetsuyoshi Kariya, Nobuyuki Hamajima.

**Methodology:** Yu Mon Saw, Thu Nandar Saw, Thet Mon Than, Moe Khaing, Pa Pa Soe, San Oo, Ei Mon Win, Etsuko Fuchita, Tetsuyoshi Kariya, Shigemi Iriyama, Nobuyuki Hamajima.

**Project administration:** Yu Mon Saw, Thu Nandar Saw, Thet Mon Than, Moe Khaing, Pa Pa Soe, San Oo, Su Myat Cho, Ei Mon Win, Aye Myat Mon, Tetsuyoshi Kariya, Nobuyuki Hamajima.

**Resources:** Yu Mon Saw.

**Supervision:** Yu Mon Saw, Moe Khaing, San Oo, Shigemi Iriyama, Nobuyuki Hamajima.

**Validation:** Yu Mon Saw, Thu Nandar Saw, Thet Mon Than, Moe Khaing, Pa Pa Soe, San Oo, Su Myat Cho, Ei Mon Win, Aye Myat Mon, Etsuko Fuchita, Tetsuyoshi Kariya, Shigemi Iriyama, Nobuyuki Hamajima.

**Visualization:** Yu Mon Saw, Thu Nandar Saw, Thet Mon Than, Moe Khaing, Pa Pa Soe, San Oo, Su Myat Cho, Ei Mon Win, Aye Myat Mon, Etsuko Fuchita, Tetsuyoshi Kariya, Shigemi Iriyama, Nobuyuki Hamajima.

**Writing – original draft:** Yu Mon Saw, Thu Nandar Saw, Thet Mon Than, Moe Khaing.

**Writing – review & editing:** Yu Mon Saw, Nobuyuki Hamajima.

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
