## [Decision Letter · Decision Letter 0]

27 Apr 2020

PONE-D-20-03550

Cognitive impairment and its risk factors among Myanmar elderly using the Revised Hasegawa’s Dementia Scale: A cross-sectional study in Nay Pyi Taw, Myanmar

PLOS ONE

Dear Dr. Saw

Thank you for submitting your manuscript to PLOS ONE. After careful consideration, we feel that it has merit but does not fully meet PLOS ONE’s publication criteria as it currently stands. Therefore, we invite you to submit a revised version of the manuscript that addresses all the points raised during the review process.

We would appreciate receiving your revised manuscript by Jun 11 2020 11:59PM. To enhance the reproducibility of your results, we recommend that if applicable you deposit your laboratory protocols in protocols.io, where a protocol can be assigned its own identifier (DOI) such that it can be cited independently in the future. For instructions see: http://journals.plos.org/plosone/s/submission-guidelines#loc-laboratory-protocols

We look forward to receiving your revised manuscript.

Kind regards,

Gianluigi Forloni

Academic Editor

PLOS ONE

Journal Requirements:

3. Please include a caption for figure 1.

Reviewers' comments:

Reviewer's Responses to Questions

**Comments to the Author**

1. Is the manuscript technically sound, and do the data support the conclusions?

Reviewer #1: Yes

Reviewer #2: Partly

2. Has the statistical analysis been performed appropriately and rigorously? 

Reviewer #1: Yes

Reviewer #2: Yes

3. Have the authors made all data underlying the findings in their manuscript fully available?

Reviewer #1: Yes

Reviewer #2: Yes

4. Is the manuscript presented in an intelligible fashion and written in standard English?

Reviewer #1: Yes

Reviewer #2: Yes

5. Review Comments to the Author

Reviewer #1: Summary: The current manuscript investigated the prevalence of cognitive impairment and related comorbidities among Myanmar elderly. The authors show that the prevalence of impaired cognitive functions among participants was from 23% in males to 32 % in females. Authors also evidence that cognitive impairment was associated with age, illiterate, and general health.

Although authors present exciting findings and I would read the final manuscripts, some aspects could be improved.

Please consider the following suggestions for revision:

Introduction: Overall, the introduction provides a broad background and rationale for the research. However, it lacks information on the area of the background relative to the numerous socio-demographic, physical, and mental conditions associated with cognitive impairment. Moreover, the rationale regarding the choice to use the revised Hasegawa's dementia scales for the assessment of cognitive impairment is not clear. More studies evidence that this general test does not always lead to a clear definition of the prevalence. Therefore, it would be appropriate to justify the choice. Finally, the researchers' hypotheses are unclear.

Methods: The method is succinct and comprehensive.

Analysis: the analyses are well conducted. However, it could be useful to define the confounding variables controlled with AORs.

Results: The summary of the study provided is well-defined and fits according to the analysis plan provided. Could it be useful for reporting X2 in the table?

Discussion: This section could be improved, also introducing the specific comments included below.

Some explanations about brain change in the elderly could be helpful to define better why the elderly are more vulnerable to cognitive impairment.

The explanation about BMI is not correct. Recent studies reported a higher risk of occurrence of cognitive impairment in individuals with excessive body weight and high BMI; it could be useful to report these studies and hypothesize a different explanation for this (e.g., non-linear relationship). It is reporting these results to provide a complete view.

Some aspects could be discussed, such as hypertension, behavioral risk factors (alcohol, cigarettes, etc.), and diabetes. All of these are considered as risk factors for cognitive impairment from a lot of studies (and systematic reviews). In particular, to consider that in all samples, 57% of females and 67% of the male are hypertensive.

I suggest the reading of the following systematic review:

“Favieri, F., Forte, G. & Casagrande M. (2019). The executive functions in overweight and obesity: a systematic review of neuropsychological cross-sectional and longitudinal studies. Frontiers in Psychology, 10, 1-27, DOI: 10.3389/fpsyg.2019.02126”;

“Forte, G., De Pascalis, V., Favieri, F., & Casagrande, M. (2020). Effects of Blood Pressure on Cognitive Performance: A Systematic Review. Journal of Clinical Medicine, 9(1), 34” ;

“Cuevas, H. E. (2019). Type 2 diabetes and cognitive dysfunction in minorities: a review of the literature. Ethnicity & health, 24(5), 512-526.”

“Conti, A. A., McLean, L., Tolomeo, S., Steele, J. D., & Baldacchino, A. (2019). Chronic tobacco smoking and neuropsychological impairments: A systematic review and meta-analysis. Neuroscience & Biobehavioral Reviews, 96, 143-154.”

“Dye, L., Boyle, N. B., Champ, C., & Lawton, C. (2017). The relationship between obesity and cognitive health and decline. Proceedings of the nutrition society, 76(4), 443-454”.

Regarding limitations, It could be useful reporting the percentage of participants in whom an impairment has been confirmed.

Conclusion: the conclusions appear to be a summary of the results, I suggest reporting the usefulness of this study and further perspective.

General comment: I would also encourage the authors to check all references and to proofread the manuscript to improve the English language.

Reviewer #2: 1. p. 4 - bottom: clarify what you mean by ‘unable to sum the value’.

2. P. 5: please identify how many people were approached to participate in the study, and whether people who refused participation were different from individuals who agreed to participate.

3. Please provide a theoretical justification for choosing the covariates set. Explain why you chose the existing set of covariates instead of other covariates.

4. Justify the decision to stratify the HDS-R instead of treating it as a continuous variable.

5. Cognitive function and impairment can differ across various domains such as working memory, executive function, psychomotor speed, etc. In a study to identify factors associated with cognitive impairment, the examination of specific cognitive domains should be essential. The omission of such an examination is a limitation of a study that purports to seek out risk factors for cognitive impairment.

6. Ethical considerations: please explain how you ensured that study participants with cognitive impairment were capable of providing informed consent.

7. The findings are not novel or surprising, and the choice of variables was not anchored in any sort of theory. As such, the article seemed to be a fishing expedition to find statistically significant results. This is a problem because the wide confidence intervals in the regression analyses suggest the study was underpowered to detect certain effects.

8. Please report the manuscript in accordance with the STROBE guidelines for reporting observational research.

9. Abstract methods: you conducted a ‘multivariable’, not ‘multivariate’, logistic regression analysis.

6. PLOS authors have the option to publish the peer review history of their article (what does this mean?). If published, this will include your full peer review and any attached files.

Reviewer #1: No

Reviewer #2: No

---

## [Author Response · Author response to Decision Letter 0]

4 Jun 2020

Response letter (Response to Reviewers)

PONE-D-20-03550: Cognitive impairment and its risk factors among Myanmar elderly using the Revised Hasegawa’s Dementia Scale: A cross-sectional study in Nay Pyi Taw, Myanmar

Thank you very much editor and reviewer for your valuable comments and suggestions. We have revised the manuscript according to your suggestions. The revised and edited sentences (and words) are mentioned using a track-changes function in the revised manuscript. We also submitted a clean version of revised manuscript as a separate file. In below responses, we noted reviewer’s comments in black color and our responses in blue color. 

Reviewer #1: 

Summary: The current manuscript investigated the prevalence of cognitive impairment and related comorbidities among Myanmar elderly. The authors show that the prevalence of impaired cognitive functions among participants was from 23% in males to 32 % in females. Authors also evidence that cognitive impairment was associated with age, illiterate, and general health. Although authors present exciting findings and I would read the final manuscripts, some aspects could be improved. Please consider the following suggestions for revision:

Q-1: Introduction: Overall, the introduction provides a broad background and rationale for the research. However, it lacks information on the area of the background relative to the numerous socio-demographic, physical, and mental conditions associated with cognitive impairment. Moreover, the rationale regarding the choice to use the revised Hasegawa's dementia scales for the assessment of cognitive impairment is not clear. More studies evidence that this general test does not always lead to a clear definition of the prevalence. Therefore, it would be appropriate to justify the choice. Finally, the researchers' hypotheses are unclear. 

A-1: Authors’ response: Thank you very much for the comments. As suggested, the information on the area of background relative to the numerous socio-demographic, physical, and mental conditions associated with cognitive impairment were added in the introduction section. The previous second paragraph in the introduction section, “Numerous socio-demographic, physical, and mental conditions have been found to be associated with cognitive impairment. Older age [8,9], being female [10-12], poor marital relationship [10,13-15], low educational level in earlier life [16,17], solitary living [10,13-15], low level of physical activities [10,18-21], underweight [10,22], hypertension [23], and diabetes mellitus [24] are important risk factors for cognitive decline. Meanwhile, high socio-economic status [25,26]; high level of social activities [13,27]; good nutrition [28]; being free from anxiety, stress [14], or depression [15]; as well as high level of physical activities [18–22] have been observed to be protective factors against cognitive impairment.” is revised and read as follows: “Numerous socio-demographic, physical, and mental conditions have been found to be associated with cognitive impairment. Older age [8,9], being female [10-12], poor marital relationship [10,13-15], low educational level in earlier life [16,17], solitary living [10,13-15], low level of physical activity [10,18-21], chronic tobacco smoking [22], alcohol consumption [23], obesity [24,25], visual impairment [26], hypertension [27], and diabetes mellitus [28,29] are important risk factors for cognitive decline. Meanwhile, high socioeconomic status [30,31]; high level of social activities [13,32]; good nutrition [33]; being free from anxiety, stress [14], or depression [15]; as well as high level of physical activity [18–21] have been observed to be protective factors against cognitive impairment.

Age and gender are unmodifiable risk factors for cognitive decline. In the normal aging process, brain volume shrinkage, especially in the prefrontal cortex, which is responsible for memory performance, starts after 40 years of age and a rapid decrease in brain volume has been observed in patients over 70 years of age [34]. Nowadays, the world’s population is aging as advanced medical technological advances increase life expectancy, and age-related cognitive declination has become a major issue. Non-communicable diseases (NCDs) such as hypertension, diabetes mellitus, and obesity due to low physical activity accompany aging [24,25,29,35]. These are responsible for rapid brain aging and cerebral-vascular accidents, provoking the action of pro-inflammatory cytokines with the resultant chronic inflammation and cerebral white matter atrophy leading to cognitive impairment [24,25,34]. Cognitive impairment is also influenced by hormonal changes, and females suffer most, especially after menopause, due to decreased estrogen levels [10,12]. 

 Learning or education, especially in childhood, enhances brain structure and development by increasing brain vascularization, synapse number, and connections, which improve cognitive function [36,37]. Higher education levels are associated with lower cognitive decline as learning creates favorable structures and neurochemical alterations in the brain [36,37]. High socioeconomic status, high physical and social activities, and less dependency are protective factors for cognitive impairments [13,30-32]. People with high socioeconomic status generally have more social contact and activities that make them more active, less dependent, and perform higher physical activities leading to slower cognitive decline [13,30-32]. Moreover, these people can have good nutrition and can easily access the health services they need, maintaining their health in a good state that can delay cognitive declination [30-32].

 On the contrary, elderly people living a solitary life and with failed marital status or unhealthy behaviors such as chronic smoking or alcohol consumption had a higher risk of developing cognitive impairments [10,13,15,22]. Elderly individuals who live alone and are widowed, divorced, or separated may have low social contact and activities that can initiate or exacerbate lower mood or depression, the high-risk factor for cognitive impairment [10,13-15]. Elderly people leading a lonely life may also harbor risky behaviors such as chronic alcohol drinking or smoking, as there is no family member to control them, which can increase cognitive impairment [10,13,15,22].” [Introduction, Line 63-102, Page 3-5]

Moreover, the rationale regarding the choice to use the revised Hasegawa’s dementia scale for the assessment of cognitive impairment was also added in the introduction section of the revised manuscript. The last paragraph of the previous introduction section “Few studies have been performed on the cognitive function assessment among elderly in Myanmar. The age- and sex-specific cognitive functions, as well as the influencing factors, are not yet reported. This kind of information is extremely useful to the planning of programs for elderly care, treatment, and prevention of dementia strategies. Therefore, the present study aimed to identify the rate of cognitive impairment and related comorbidities among Myanmar elderly.” was revised as follows: “Few studies have been performed on the cognitive function assessment among elderly in Myanmar. Assessing cognitive impairment among the Myanmar elderly using the Myanmar-translated version of the Revised Hasegawa’s Dementia Scale (HSR-D) is still lacking. The HSR-D Myanmar version was officially translated by Myanmar and Japanese scientists, and some modifications were made according to the local context [41]. The scale does not include questions assessing the reading and writing ability of the respondents, making it convenient to use for illiterate people and every ethnic group using different languages in Myanmar. Therefore, it can be used as a screening tool to easily detect cognitive impairment among communities even by the basic health staff, which could be quite helpful in Myanmar with limited human resources for health. Age- and sex-specific cognitive functions, as well as the influencing factors, have not yet been reported. This kind of information is extremely useful to plan programs for elderly care, treatment, and prevention of dementia strategies. Therefore, the present study aimed to identify the rate of cognitive impairment and its risk factors among the elderly in Myanmar.” [Introduction, Line 114-127, Page 5-6]

To reflect the reviewer’s comments, the previous usage “the prevalence of the cognitive impairment” revised and read as follows: “the rate of cognitive impairment” throughout the revised manuscript. 

Q-2: Methods: The method is succinct and comprehensive. Analysis: the analyses are well conducted. However, it could be useful to define the confounding variables controlled with AORs. 

A-2: Authors’ response: Thank you very much for your valuable suggestion. As suggested, we added the definition of confounding variables one of the sub-section of methods “Independent variables 

Socio-demographic characteristics, substance use behaviors, and health problems were considered as independent variables. The current age was categorized into three groups (60-69, 70-79, and ≥80) based on the 10-year age intervals. Marital status was categorized into three groups (single, married, and separated/divorced/windowed). Education was divided into three groups according to the educational background of respondents of the elderly: middle school and above, primary school, and only read and write, and illiterate. Family type was categorized into five groups: living alone, nuclear, extended, three generations, and skip generation to learn how family structures affect the cognitive functions of the elderly. 

Substance use behaviors were grouped into the following categories: non-users (never use), ex-users, occasional users, and daily users to see the effect on the cognitive functions of respondents. Self-rated health, physical activities, and vision status were divided into two categories. The nutritional status of the elderly may play an important role in impairment of cognitive function. Therefore, Body mass index (BMI) was categorized as underweight, normal, overweight, and obese. Hypertension and diabetes mellitus were grouped into two categories according to self-reported and measurement results. The measurement cutoff point of blood pressure was 140/90 mmHg (hypertension: ≥140/90 mmHg) and random blood sugar was 200 mg/dL (diabetes mellitus: ≥ 200 mg/dL).” [Methods, Line 114-127, Page 7-8]

In addition, we also amended the previous footnote of Table 4 “Adjusted for the variables listed in the table.” to read as follows: “Adjusted for age, gender, marital status, education, dependent, family type, alcohol, smoking and smokeless tobacco use, self-rated health, no. of” comorbidity, low physical activities, vision status, BMI, hypertension, and diabetes mellitus. [Results-Table 4, Line 33-36, Page 16]

Q-3: Results: The summary of the study provided is well-defined and fits according to the analysis plan provided. Could it be useful for reporting X2 in the table?

A-3: Authors’ response: Thank you very much for your valuable suggestion. As suggested, we reported Pearson's chi-square test value in the table 1 and 2. [Results Table 1 and 2, Line 217 and 230, Page 11-12]

Table 1 Socio-demographic characteristic of participants 

Characteristics Total (N=757) Male (n=246) Female (n=511) X2 ‡

 N % n % n % 

Age 15.01*

 60-64 203 26.8 51 20.7 152 29.7 

 65-69 185 24.4 63 25.6 122 23.9 

 70-74 170 22.5 65 24.4 105 20.5 

 75-79 95 12.5 24 9.8 71 13.9 

 80-84 68 9.0 26 106 42 8.3 

 85-89 27 3.6 12 4.9 15 2.9 

 ≥ 90 9 1.2 5 2.0 4 0.8 

Education 88.91***

 Illiterate 165 21.8 11 4.5 154 30.1 

 Primary school 487 64.3 173 70.3 314 61.4 

 Middle school 72 9.5 41 16.7 31 6.1 

 High school 22 2.9 16 6.5 6 1.2 

 University and above 11 1.5 5 20.0 6 1.2 

Marital status 75.94***

 Single 49 6.5 13 5.3 36 7.0 

 Married 393 51.9 183 74.4 210 41.1 

 Others 315 41.6 50 20.3 265 51.9 

Family type 12.43*

 Living alone 56 7.5 10 4.1 46 9.0 

 Nuclear 108 14.3 47 19.1 61 11.9 

 Extended 438 57.8 144 58.5 294 57.5 

 Three generation 113 14.9 32 13.0 81 15.9 

 Skip generation 42 5.5 13 5.3 29 5.7 

Low physical activities 4.50*

 No 232 69.4 158 64.2 367 71.8 

 Yes 525 30.6 88 35.8 144 28.2 

Place of interview 10.33

 Ngan Sat RHC 60 7.9 26 10.6 34 6.7 

 Tha Pyay Pin RHC 97 12.8 26 10.6 71 13.9 

 Zee Kone RHC 81 10.7 29 11.8 52 10.2 

 Nat Tha Ye RHC 92 12.2 30 12.2 62 12.1 

 Taung Po Thar RHC 54 7.2 18 7.3 36 7.0 

 Ma Dot Pin RHC 45 5.9 12 4.9 33 6.5 

 Baw Di Gone RHC 60 7.9 19 7.7 41 8.0 

 Tha Wut Hti RHC 70 9.2 27 11.0 43 8.4 

 Nyaung Lont RHC 87 11.5 31 12.6 56 11.0 

 Pyi San Aung RHC 49 6.5 11 4.5 38 7.4 

 Si Pin Thar Yar RHC 62 8.2 17 6.8 45 8.8 

RHC: Rural Health Center. ‡Pearson's chi-square test. *p<0.05, **p<0.01, ***p<0.001

Table 2 Substance use behaviors and health-related characteristic of participants 

Characteristics Total (N=757) Male (n=246) Female (n=511) X2‡

 N % n % n % 

Smoking 50.69***

 Never smoke 516 68.2 125 50.8 391 76.5 

 Ex-smoker 102 13.5 50 20.3 52 10.2 

 Occasional smoker 34 4.5 17 6.9 17 3.3 

 Daily smoker 105 13.8 54 22.0 51 10.0 

Smokeless tobacco use 9.40*

 Never use 457 60.4 139 56.5 318 62.2 

 Ex-user 27 3.6 15 6.1 12 2.3 

 Occasional user 76 10.0 21 8.5 55 10.8 

 Daily user 197 26.0 71 28.9 126 24.7 

Alcohol drinking 119.37***

 Never Drink 684 90.4 181 73.6 503 98.4 

 Ex-drinker 52 6.9 44 17.9 8 1.6 

 Occasional/social drinker 17 2.2 17 6.9 0 0 

 Heavy drinker 4 0.5 4 1.6 0 0 

Self-rated health 4.50

 Very poor/poor 267 35.3 76 30.9 191 37.4 

 Fair 237 31.3 76 30.9 161 31.5 

 Good/very good 253 33.4 94 38.2 159 31.1 

Comorbidity 5.81

 No disease 151 19.9 54 22.0 97 19.0 

 At least one disease 342 45.2 121 49.2 221 43.2 

 Two or more diseases 264 34.9 71 28.8 193 37.8 

Vision status 2.74

 Good 371 49.0 128 52.0 243 47.6 

 Fair 331 43.7 105 42.7 226 44.2 

 Poor 55 7.3 13 5.3 42 8.2 

BMI 5.72

 Under weight (11.9-18.4) 220 29.1 69 28.0 151 29.5 

 Normal (18.5-22.9) 283 37.3 106 43.1 177 34.6 

 Overweight (23.0-24.9) 106 14.0 29 11.8 77 15.2 

 Obese (=>25) 148 19.6 42 17.1 106 20.7 

Hypertension 7.75*

 No 297 39.2 79 32.1 218 42.7 

 Yes 460 60.8 167 67.9 293 57.3 

Diabetes mellitus 0.20

 No 599 79.1 197 80.1 402 78.7 

 Yes 158 20.9 49 19.9 109 21.3 

‡Pearson's chi-square test. *p<0.05, **p<0.01, ***p<0.001

Q-4: Discussion: This section could be improved, also introducing the specific comments included below. Some explanations about brain change in the elderly could be helpful to define better why the elderly are more vulnerable to cognitive impairment.

A-4: Authors’ response: Thank you very much for the comments. As suggested, we revised the pervious sentences “In this study, the participants who were older than 70 years had a higher risk for developing cognitive impairment compared with the 60–69 years old age group. This finding is consistent with other studies [8,9,23,24]. The rate of cognitive impairment is the highest in the age group of 85 years and older, ranging from 16.7% in China [8] to 43% in Germany [9]. Studies have also estimated that elderly aged 75 years or older account for 80% of patients with dementia [23,24]. Older age is the greatest unmodifiable risk factor for cognitive impairment [16], and should be taken into consideration in cognitive impairment prevention, as life expectancy is increasing worldwide [16]. Aging can also combine with other comorbidities and can exacerbate the situation [16]. As such, health policymakers and stakeholders in Myanmar should initiate community preventive measures or strategies for cognitive impairment as early as possible to mitigate its adverse consequences.” to add more explanations about brain change in the elderly and read as follows: “In this study, the participants older than 70 years had a higher risk of developing cognitive impairment compared with the 60–69 years old age group. This finding is consistent with those of other studies [8,9,27,28]. The rate of cognitive impairment is the highest in the age group of 85 years and older, ranging from 16.7% in China [8] to 43% in Germany [9]. Studies have also estimated that elderly aged 75 years or older account for 80% of patients with dementia [27,28]. Aging is associated with several changes in brain structure and function. Brain volume shrinkage started around or after 40 years, and the shrinkage rate increased especially for those over 70, even in the normal aging process [34]. The most affected area is the prefrontal cortex, which is responsible for memory performance. Reduction in cortical volume associated with increased white matter lesions in the elderly leads to executive function declination and cognitive impairment [34]. Moreover, aging is usually associated with NCDs such as hypertension and type 2 diabetes mellitus [29,35]. Hypertension and diabetes mellitus account for small or large vascular changes leading to cerebral-vascular accidents, strokes, cerebral hemorrhage or micro-cerebral infarcts. As a result, cognitive function impairment is inevitable due to brain volume reduction [29,34,35]. Older age is the greatest unmodifiable risk factor for cognitive impairment [16], and should be taken into consideration in cognitive impairment prevention, as life expectancy is increasing worldwide. Aging can also combine with other comorbidities and can exacerbate the situation [16]. As such, health policymakers and stakeholders in Myanmar should initiate community preventive measures or strategies for cognitive impairment as early as possible to mitigate its adverse consequences.” [Discussion, Line 287-307, Page 17]

 Furthermore, the brain changes associated with obesity (BMI and cognitive impairment) was mentioned in the revised paragraph of discussion section as follows: “Moreover, middle-age obesity is more likely to be associated with rapid brain ageing and cerebral white matter atrophy due to the action of pro-inflammatory cytokines causing chronic inflammation and metabolic diseases [23,24,32].” [Discussion, Line 377-379, Page 20]

Q-5: The explanation about BMI is not correct. Recent studies reported a higher risk of occurrence of cognitive impairment in individuals with excessive body weight and high BMI; it could be useful to report these studies and hypothesize a different explanation for this (e.g., non-linear relationship). It is reporting these results to provide a complete view.

Some aspects could be discussed, such as hypertension, behavioral risk factors (alcohol, cigarettes, etc.), and diabetes. All of these are considered as risk factors for cognitive impairment from a lot of studies (and systematic reviews). In particular, to consider that in all samples, 57% of females and 67% of the male are hypertensive.

I suggest the reading of the following systematic review:

“Favieri, F., Forte, G. & Casagrande M. (2019). The executive functions in overweight and obesity: a systematic review of neuropsychological cross-sectional and longitudinal studies. Frontiers in Psychology, 10, 1-27, DOI: 10.3389/fpsyg.2019.02126”;

“Forte, G., De Pascalis, V., Favieri, F., & Casagrande, M. (2020). Effects of Blood Pressure on Cognitive Performance: A Systematic Review. Journal of Clinical Medicine, 9(1), 34” ;

“Cuevas, H. E. (2019). Type 2 diabetes and cognitive dysfunction in minorities: a review of the literature. Ethnicity & health, 24(5), 512-526.”

“Conti, A. A., McLean, L., Tolomeo, S., Steele, J. D., & Baldacchino, A. (2019). Chronic tobacco smoking and neuropsychological impairments: A systematic review and meta-analysis. Neuroscience & Biobehavioral Reviews, 96, 143-154.”

“Dye, L., Boyle, N. B., Champ, C., & Lawton, C. (2017). The relationship between obesity and cognitive health and decline. Proceedings of the nutrition society, 76(4), 443-454”.

A-5: Authors’ response: Thank you for your comments. The systematic reviews mentioned by the reviewer were thoroughly read and cited in the revised manuscript. As suggested, the previous paragraph explaining about obesity in the discussion section “Among all participants, those whose BMI was in the normal or obese category were observed to have a lower risk for developing cognitive impairment. A high BMI reflects good nutrition [40] as well as high body fat, which is favorable for increased glucose metabolism, especially cerebral glucose metabolism, which enhanced cognitive performance [41,42]. Studies have also observed that individuals with a low BMI have a higher risk for developing cognitive impairment compared with those with a high BMI [10,22].” was revised and some aspects regarding hypertension, behavioral risk factors (alcohol, cigarettes, etc.), and diabetes were added and to read as follows: “Participants whose BMI was in the normal or obese category were observed to have a lower risk of developing cognitive impairment in this study. Many studies agree that maintaining normal body weight throughout the life span is considered to be a protective factor for cognitive impairment, which is consistent with this study [24,25]. Increased BMI associated with central obesity in the middle years of life has been observed as one of the risk factors for cognitive decline in older age by recent systematic reviews and meta-analyses [24,25]. Obesity in the middle years of life is usually related to hypertension, stroke, diabetes mellitus, and dyslipidemia [24,25,29,35]. Moreover, middle-age obesity is more likely to be associated with rapid brain aging and cerebral white matter atrophy due to the action of pro-inflammatory cytokines causing chronic inflammation and metabolic diseases [24,25,34]. Pathophysiological changes in obesity are considered to be related mainly to adiposity distribution, which cannot be measured directly by BMI as it fails to differentiate muscles from adipose tissues [24,25]. This study used BMI to categorize underweight, normal, overweight, and obese among the participants. This is one of the possible reasons why obese participants in this study had a lower risk of cognitive impairment. Another reason for the lower risk of cognitive impairment among obese participants in this study is that their obesity may start in their late-life as obesity in the later years of life (over 76 years of age) is found to be associated with slower cognitive decline in some studies [24,25].

In this study, 13.8% of the participants were daily smokers and 0.5% were heavy drinkers. Chronic smokers were more likely to be alcohol drinkers, and chronic tobacco smoking is associated with cognitive decline and the development of neurocognitive diseases in later life [22]. However, this association was not found in this study. This may be due to the limited number of study participants and the study conducted in the same demographic area in Myanmar.” [Discussion, Line 370-393, Page 20-21]

Q-6: Regarding limitations, it could be useful reporting the percentage of participants in whom an impairment has been confirmed. Conclusion: the conclusions appear to be a summary of the results, I suggest reporting the usefulness of this study and further perspective.

A-6: Authors’ response: Thank you very much for your comments. As suggested, the percentage of the participants with cognitive impairment was reported in the limitations. The previous limitations sentence, “This study has a number of strengths and limitations. This is the first study to investigate the rate of impaired cognitive function using the Myanmar-translated version of HDS-R and its related comorbidities among Myanmar elderly. HDS-R has its own advantages. As it does not include questions assessing the reading and writing ability of respondents, it is convenient to use in illiterate people or in minor ethnic groups who cannot read and write the Myanmar language. Cognitive impairment could be checked easily by basic health staff using HDS-R. However, cognitive impairment among the participants in this study was not confirmed by a psychiatrist. The study was conducted in one region in the central part of Myanmar, which limited the generalizability of the results, given that different socio-demographic features are represented across Myanmar. The causal relation between cognitive impairment and different risk factors could not be explored clearly owing to the cross-sectional design of the study.” was revised as follows: “This study has several strengths and limitations. This is the first study to investigate the prevalence of impaired cognitive function using the Myanmar-translated version of the HDS-R and its related comorbidities among Myanmar elderly. It was observed that 23.6% of males and 32.9% of females in this study had cognitive impairment detected by the HDS-R. The HDS-R has its own advantages. As it does not include questions assessing the reading and writing ability of respondents, it is convenient to use in illiterate people or in minor ethnic groups who cannot read and write the Myanmar language. Cognitive impairment could be checked easily by basic health staff using the HDS-R. However, cognitive impairment among the participants in this study was not confirmed by a psychiatrist; therefore, cognitive impairment in specific domains such as executive function, spatial working memory, processing speed, attention, verbal memory, or verbal fluency could not be ruled out. The study was conducted in one region in the central part of Myanmar, which limited the generalizability of the results, given that different socio-demographic features are represented across Myanmar. The causal relationship between cognitive impairment and different risk factors could not be explored clearly owing to the cross-sectional design of the study.” [Discussion, Line 394-408, Page 21]

According to the reviewer’s comments, the usefulness of this study and further perspective was reported in the conclusion of the revised manuscript. The previous paragraph in the conclusion section “About 30% of the participants had cognitive impairment and female participants were significantly more likely to develop cognitive impairment compared to male. Being over 70 years old, a low educational level, dependency, solitary living, and poor self-rated health were associated with a higher risk for developing cognitive impairment. Meanwhile, living with one’s family and good self-rated health were protective factors. Policymakers need to consider implementing community preventive measures or strategies regarding cognitive impairment as early as possible to mitigate its adverse consequences.” was revised as follows: “Three out of five elderly participants reported having cognitive impairment and female participants were significantly more likely to develop cognitive impairment. Being over 70 years old, a low educational level, dependency, solitary living, and poor self-rated health were associated with a higher risk of developing cognitive impairment. Meanwhile, living with one’s family and having good self-rated health were protective factors. Based on scientific evidence, policymakers need to consider implementing community preventive measures or strategies regarding cognitive impairment and gender differences as early as possible to mitigate its adverse consequences among Myanmar elderly. Screening for cognitive impairment using the Myanmar language version of the HDS-R should be confirmed by the clinical diagnosis in the further studies so that even the basic health staff can screen for cognitive impairment among the general population at the most basic lever. Therefore, it could be helpful in limited health workforce settings. Future research should be performed not only to detect general cognitive impairment but also to differentiate specific cognitive domains impairments among the Myanmar elderly. Longitudinal studies are needed to observe the causal and protective factors associated with cognitive impairments and associated comorbidities in Myanmar.” [Conclusion: Line 410-424, Page 22]

Furthermore, the previous conclusion section in the abstract “Conclusion: About 30% of participants had cognitive impairment, and female participants were significantly more likely to develop cognitive impairment compared to male participants. Being over 70 years old, having a low educational level, dependency, solitary living, and poor self-rated health were associated with a higher risk for cognitive impairment. Meanwhile, living with families and self-reported good health were protective factors against cognitive impairment.” was also revised into as follows: “Conclusion: Using the HDS-R Myanmar version, this study reported that there out of five elderly participants had cognitive impairment, and its risk factors, altering policy makers that Myanmar needs to prepare for adequate healthcare services and social support for elderly with cognitive impairment. Future research should be performed not only to detect general cognitive impairment but also to differentiate specific cognitive domains impairments among Myanmar elderly. Longitudinal studies are needed to observe the causal and protective factors associated with cognitive impairments in Myanmar.” [Abstract, Line 44-50, Page 2]

Q-7: General comment: I would also encourage the authors to check all references and to proofread the manuscript to improve the English language.

A-7: Authors’ response: Thank you very much for your comments. As suggested, we checked all the references and revised it accordingly. Our manuscript is proofreader by the professional proofreaders with public health experience to improve the language quality of the revised manuscript.

Reviewer #2: 

Q-1: p. 4 - bottom: clarify what you mean by ‘unable to sum the value’.

A-1: Authors’ response: Thank you for your question. We corrected the previous word ‘unable to sum the value’ into ‘unable to subtract the value’. We also revised the previous sentence from Materials and Methods section “Elderly individuals who lived in the study area less than six months, who were diagnosed with cognitive impairments along with mental and physical disorders (seriously ill), and who did not understand the Myanmar language and were unable to sum the value were excluded from the study.” to read as follows: “Elderly individuals who lived in the study area less than six months, who were diagnosed with cognitive impairments along with mental and physical disorders (seriously ill), and who did not understand the Myanmar language and were unable to subtract the value were excluded from the study.” [Methods, Line 135-138, Page 6]

Q-2: P. 5: please identify how many people were approached to participate in the study, and whether people who refused participation were different from individuals who agreed to participate.

A-2: Authors’ response: In total, 971 elderly people approached to participate in this study. Elderly individuals who lived in the study area less than six months, who were diagnosed with cognitive impairments along with mental and physical disorders (seriously ill), and who did not understand the Myanmar language and were unable to subtract the value were excluded. As the data collection was collected at rural health centers, the elderly people who would like to voluntarily participate this study came to the rural health centers. The elderly who lived quite far from rural health centers with transportation difficulties, those who were out of town at the time of data collection, and those who had to attend their personal or familial events at the time of data collection refused to participate in this study. As suggested, we added response rate of the survey as follows: “In total, 971 elderly people were invited to participate in this survey. Of them, 811 elderly participants (males: 264 [32.6%], females: 547 [67.4%]) provided written informed consent and agreed to participate in this study. The response rate was 92.5%.” [Methods, Line 150-155, Page 7]

Q-3: Please provide a theoretical justification for choosing the covariates set. Explain why you chose the existing set of covariates instead of other covariates.

A-3: Authors’ response: Thanks for your suggestions. We chose the covariates in our study after thorough literature reviews reporting the risk factors for cognitive impairment, and found those covariates were applicable to study design, local context and the community setting of this study. As suggested, the theoretical justification for choosing the covariates set was mentioned in the revised manuscript as follows: “Numerous socio-demographic, physical, and mental conditions have been found to be associated with cognitive impairment. Older age [8,9], being female [10-12], poor marital relationship [10,13-15], low educational level in earlier life [16,17], solitary living [10,13-15], low level of physical activity [10,18-21], chronic tobacco smoking [22], alcohol consumption [23], obesity [24,25], visual impairment [26], hypertension [27], and diabetes mellitus [28,29] are important risk factors for cognitive decline. Meanwhile, high socioeconomic status [30,31]; high level of social activities [13,32]; good nutrition [33]; being free from anxiety, stress [14], or depression [15]; as well as high level of physical activity [18–21] have been observed to be protective factors against cognitive impairment.

Age and gender are unmodifiable risk factors for cognitive decline. In the normal aging process, brain volume shrinkage, especially in the prefrontal cortex, which is responsible for memory performance, starts after 40 years of age and a rapid decrease in brain volume has been observed in patients over 70 years of age [34]. Nowadays, the world’s population is aging as advanced medical technological advances increase life expectancy, and age-related cognitive declination has become a major issue. Non-communicable diseases (NCDs) such as hypertension, diabetes mellitus, and obesity due to low physical activity accompany aging [24,25,29,35]. These are responsible for rapid brain aging and cerebral-vascular accidents, provoking the action of pro-inflammatory cytokines with the resultant chronic inflammation and cerebral white matter atrophy leading to cognitive impairment [24,25,34]. Cognitive impairment is also influenced by hormonal changes, and females suffer most, especially after menopause, due to decreased estrogen levels [10,12]. 

 Learning or education, especially in childhood, enhances brain structure and development by increasing brain vascularization, synapse number, and connections, which improve cognitive function [36,37]. Higher education levels are associated with lower cognitive decline as learning creates favorable structures and neurochemical alterations in the brain [36,37]. High socioeconomic status, high physical and social activities, and less dependency are protective factors for cognitive impairments [13,30-32]. People with high socioeconomic status generally have more social contact and activities that make them more active, less dependent, and perform higher physical activities leading to slower cognitive decline [13,30-32]. Moreover, these people can have good nutrition and can easily access the health services they need, maintaining their health in a good state that can delay cognitive declination [30-32].

 On the contrary, elderly people living a solitary life and with failed marital status or unhealthy behaviors such as chronic smoking or alcohol consumption had a higher risk of developing cognitive impairments [10,13,15,22]. Elderly individuals who live alone and are widowed, divorced, or separated may have low social contact and activities that can initiate or exacerbate lower mood or depression, the high-risk factor for cognitive impairment [10,13-15]. Elderly people leading a lonely life may also harbor risky behaviors such as chronic alcohol drinking or smoking, as there is no family member to control them, which can increase cognitive impairment [10,13,15,22].” [Introduction, Line 63-102, Page 3-5]

Q-4: Justify the decision to stratify the HDS-R instead of treating it as a continuous variable.

A-4: Authors’ response: Thank you for your question. We treated HDS-R as continuous variable because of the following literature, which stating as follows: “The most common application of the HDS-R is its use as a screening test for dementia. Using the cut-off point of 20/21, we obtained the sensitivity of 0.90 and the specificity of 0.82 in our subject. Those optimum sensitivity and specificity were achieved by regarding a score of 20 or less as suggestive of dementia”. Reference: Imai Y, Hasegawa K. The revised Hasegawa’s Dementia Scale (HDS-R)- Evaluation of its usefulness as a screening test for dementia. J Hong Kong Coll Psychiatr. 1994;4:20-24. Available: https://easap.asia/index.php/find-issues/past-issue/item/503-v4n2-9402-p20-24. 

Q-5: Cognitive function and impairment can differ across various domains such as working memory, executive function, psychomotor speed, etc. In a study to identify factors associated with cognitive impairment, the examination of specific cognitive domains should be essential. The omission of such an examination is a limitation of a study that purports to seek out risk factors for cognitive impairment.

A-5: Authors’ response: Thank you very much for your comments. As suggested, the examination of specific cognitive domains should be essential in a study to identify factors associated with cognitive impairment but this study was limited to do so. Therefore, this limitation was added and the previous paragraph of limitation “This study has a number of strengths and limitations. This is the first study to investigate the rate of impaired cognitive function using the Myanmar-translated version of HDS-R and its related comorbidities among Myanmar elderly. HDS-R has its own advantages. As it does not include questions assessing the reading and writing ability of respondents, it is convenient to use in illiterate people or in minor ethnic groups who cannot read and write the Myanmar language. Cognitive impairment could be checked easily by basic health staff using HDS-R. However, cognitive impairment among the participants in this study was not confirmed by a psychiatrist. The study was conducted in one region in the central part of Myanmar, which limited the generalizability of the results, given that different socio-demographic features are represented across Myanmar. The causal relation between cognitive impairment and different risk factors could not be explored clearly owing to the cross-sectional design of the study.” was revised to read as follows: “This study has several strengths and limitations. This is the first study to investigate the prevalence of impaired cognitive function using the Myanmar-translated version of the HDS-R and its related comorbidities among Myanmar elderly. It was observed that 23.6% of males and 32.9% of females in this study had cognitive impairment detected by the HDS-R. The HDS-R has its own advantages. As it does not include questions assessing the reading and writing ability of respondents, it is convenient to use in illiterate people or in minor ethnic groups who cannot read and write the Myanmar language. Cognitive impairment could be checked easily by basic health staff using the HDS-R. However, cognitive impairment among the participants in this study was not confirmed by a psychiatrist; therefore, cognitive impairment in specific domains such as executive function, spatial working memory, processing speed, attention, verbal memory, or verbal fluency could not be ruled out. The study was conducted in one region in the central part of Myanmar, which limited the generalizability of the results, given that different socio-demographic features are represented across Myanmar. The causal relationship between cognitive impairment and different risk factors could not be explored clearly owing to the cross-sectional design of the study.” [Discussion, Line 394-408, Page 21]

Q-6: Ethical considerations: please explain how you ensured that study participants with cognitive impairment were capable of providing informed consent.

A-6: Authors’ response: Authors’ response: Thank you for your question. If the patient is deemed incompetent to consent, consent based on legal proxies or advance directives were obtained. We revised the previous sentence from Methods section, “Research team members helped illiterate participants read the informed consent form. These participants were requested to mark their fingerprint if they understood the content of the informed consent form and agreed to participate in the study.” to read as follows: “Research team members helped illiterate participants read the informed consent form. These participants were requested to mark their fingerprint if they understood the content of the informed consent form and agreed to participate in the study. If participants were incompetent to consent, consent was taken from their legal proxies or advance directives.” [Methods, Line 200-204, Page 9]

Q-7: The findings are not novel or surprising, and the choice of variables was not anchored in any sort of theory. As such, the article seemed to be a fishing expedition to find statistically significant results. This is a problem because the wide confidence intervals in the regression analyses suggest the study was underpowered to detect certain effects.

A-7: Authors’ response: Thank you for your comment. I think you are pointing out the education variable confidence intervals in the regression analyses for UOR results “Illiterate (UOR = 14.2; 95% CI: 6.45–31.08) and AOR results “Illiterate (AOR = 9.1; 95% CI: 3.82–21.51)”. Following your suggestion, we categorized education variable to two categories (Middle school and above vs. Only read and write/primary school/illiterate) instead of three (Middle school and above, Only read and write/primary school, and Illiterate) and re-analyzed it. We found that the confidence intervals in the regression analyses becomes narrow: UOR results “Illiterate (UOR = 6.1; 95% CI: 2.91–12.75) and AOR results “Illiterate (AOR = 3.9; 95% CI: 1.77–8.42)”. The re-categorization of variable hasn’t effect the results of other variables. Other AOR results are remained the similar to our pervious analysis. Please refer below “Re-analysis Table 4”

Re-analysis Table 4 Multivariable logistic regression analysis of factors associated with cognitive impairment among Myanmar elderly (N=757)

Characteristics OR 95% CI AOR† 95% CI

Age 

 60-69 

 70-79 2.3 (1.63-3.33)*** 1.8 (1.19-2.70)**

 ≥80 4.9 (3.07-7.72)*** 3.9 (2.25-6.76)***

Gender 

 Male 

 Female 1.6 (1.12-2.25)** 1.1 (0.69-1.73)

Marital status 

 Single 

 Married 1.1 (0.53-2.31) 1.1 (0.48-2.46)

 Separated/Divorced/Windowed 2.7 (1.30-5.61)** 1.4 (0.63-3.27)

Education 

 Middle school and above 

 Only read and write /Primary school 4.4 (2.07-9.24)*** 3.4 (1.56-7.52)**

 Illiterate 14.2 (6.45-31.08)*** 9.1 (3.82-21.51)***

Dependent 

 No 

 Yes 2.5 (1.78-3.63)*** 1.6 (1.04-2.44)*

Family type 

 Living alone 

 Nuclear 0.3 (0.13-0.52)*** 0.4 (0.18-0.97)*

 Extended 0.4 (0.25-0.76)** 0.5 (0.27-0.97)*

 Three generation 0.4 (0.20-0.77)** 0.4 (0.21-0.94)*

 Skip generation 0.4 (0.17-0.94)* 0.6 (0.22-1.45)

Alcohol, smoking and smokeless tobacco use 

 Non-user (Never use) 

 Ex-user 1.4 (1.01-1.95)* 1.3 (0.89-1.88)

 Occasional user 1.6 (0.88-2.89) 1.6 (0.81-3.30)

 Daily user 1.1 (0.27-4.02) 1.2 (0.25-5.47)

Self-rated health 

 Very poor/poor/fair 

 Good/very good 0.7 (0.53-1.03) 0.7 (0.44-0.99)*

Comorbidity 

 No. diseases 

 At least one disease 1.0 (0.65-1.49) 0.8 (0.49-1.34)

 Two or more diseases 0.9 (0.61-1.45) 0.9 (0.50-1.57)

Low physical activities 

 No 

 Yes 0.9 (0.66-1.29) 1.3 (0.88-1.90)

Vision status 

 Good 

 Fair/poor 1.2 (0.85-1.58) 0.8 (0.58-1.21)

BMI § 

 Underweight 

 Normal 0.7 (0.45-0.95)* 0.9 (0.60-1.41)

 Overweigh 0.8 (0.51-1.35) 1.4 (0.77-2.24)

 Obese 0.4 (0.26-0.70)* 0.8 (0.44-1.40)

Hypertension 

 No 

 Yes 0.7 (0.54-1.02) 0.9 (0.58-1.29)

Diabetes mellitus 

 No 

 Yes 0.8 (0.55-1.21) 0.9 (0.57-1.41)

*p<0.05, **p<0.01, ***p<0.001; §BMI: Underweight (11.9-18.4), Normal (18.5-22.9), Overweigh (23.0-24.9), and Obese (≥25). † Adjusted for age, gender, marital status, education, dependent, family type, alcohol, smoking and smokeless tobacco use, self-rated health, no. of comorbidity, low physical activities, vision status, BMI, hypertension, and diabetes mellitus.”

However, our team are more interested to see how educational background of elderly effect the cognitive impairment especially among Myanmar elderly those who live in rural areas. Based on our findings, the policy makers can consider an appropriate intervention program for illiterate elderly population in rural areas those who are not getting much attention. More importantly, this study is the very first study reporting rate of cognitive impairments and its risks factors from Myanmar, a developing country with limited health resources. The previous studies also reported that low educational level in earlier life [16,17] was one of the important risk factors for cognitive decline among elderly. Regard to measurements, we carefully construct the survey questionnaire and chose variables based on literature review and reflecting the current situation rural elderly and applicable to the local community setting. To make it clear, we newly added independent variables categorization in the methods section as follow: “Independent variables 

 Socio-demographic characteristics, substance use behaviors, and health problems were considered as independent variables. The current age was categorized into three groups (60-69, 70-79, and ≥80) based on the 10-year age intervals. Marital status was categorized into three groups (single, married, and separated/divorced/windowed). Education was divided into three groups according to the educational background of respondents of the elderly: middle school and above, primary school, and only read and write, and illiterate. Family type was categorized into five groups: living alone, nuclear, extended, three generations, and skip generation to learn how family structures affect the cognitive functions of the elderly. 

Substance use behaviors were grouped into the following categories: non-users (never use), ex-users, occasional users, and daily users to see the effect on the cognitive functions of respondents. Self-rated health, physical activities, and vision status were divided into two categories. The nutritional status of the elderly may play an important role in impairment of cognitive function. Therefore, Body mass index (BMI) was categorized as underweight, normal, overweight, and obese. Hypertension and diabetes mellitus were grouped into two categories according to self-reported and measurement results. The measurement cutoff point of blood pressure was 140/90 mmHg (hypertension: ≥140/90 mmHg) and random blood sugar was 200 mg/dL (diabetes mellitus: ≥ 200 mg/dL).” [Methods, Line 114-127, Page 7-8]

Q-8: Please report the manuscript in accordance with the STROBE guidelines for reporting observational research.

A-8: Authors’ response: Thank you for your comment. We reported the STROBE guideline for reporting observation research as follow:

STROBE Statement—Checklist of items that should be included in reports of cross-sectional studies

 Item no. Recommendation Page no. Line no.

Title and abstract 1 (a) Indicate the study’s design with a commonly used term in the title or the abstract 1

2 1-3

26-50

 (b) Provide in the abstract an informative and balanced summary of what was done and what was found 2 26-50

Introduction 

Background/rationale 2 Explain the scientific background and rationale for the investigation being reported 3-5 63-113

Objectives 3 State specific objectives, including any prespecified hypotheses 5-6 114-127

Methods 

Study design 4 Present key elements of study design early in the paper 6 131

Setting 5 Describe the setting, locations, and relevant dates, including periods of recruitment, exposure, follow-up, and data collection 6-7 131-155

Participants 6 (a) Give the eligibility criteria, and the sources and methods of selection of participants 6 135-138

Variables 7 Clearly define all outcomes, exposures, predictors, potential confounders, and effect modifiers. Give diagnostic criteria, if applicable 7-8 156-186

Data sources/ measurement 8* For each variable of interest, give sources of data and details of methods of assessment (measurement). Describe comparability of assessment methods if there is more than one group 6

7

 145-151

158-168

Bias 9 Describe any efforts to address potential sources of bias 21 394-408

Study size 10 Explain how the study size was arrived at 7 151-155

Quantitative variables 11 Explain how quantitative variables were handled in the analyses. If applicable, describe which groupings were chosen and why 7-8 170- 168

Statistical methods 12 (a) Describe all statistical methods, including those used to control for confounding 8 190-193

 (b) Describe any methods used to examine subgroups and interactions - -

 (c) Explain how missing data were addressed 7 153-155

 (d) If applicable, describe analytical methods taking account of sampling strategy - -

 (e) Describe any sensitivity analyses - -

Results

Participants 13* (a) Report numbers of individuals at each stage of study—eg numbers potentially eligible, examined for eligibility, confirmed eligible, included in the study, completing follow-up, and analysed 7 150-155

 (b) Give reasons for non-participation at each stage 7 152-153

 (c) Consider use of a flow diagram - -

Descriptive data 14* (a) Give characteristics of study participants (eg demographic, clinical, social) and information on exposures and potential confounders 9-12 209-239

Table 1 Table 2

Table 3

 (b) Indicate number of participants with missing data for each variable of interest 7 153-155

Outcome data 15* Report numbers of outcome events or summary measures 12 233-239

Table 3

Main results 16 (a) Give unadjusted estimates and, if applicable, confounder-adjusted estimates and their precision (eg, 95% confidence interval). Make clear which confounders were adjusted for and why they were included 14-16 248-275

Table 4

 (b) Report category boundaries when continuous variables were categorized 7 158-168

 (c) If relevant, consider translating estimates of relative risk into absolute risk for a meaningful time period - -

Other analyses 17 Report other analyses done—eg analyses of subgroups and interactions, and sensitivity analyses 

Discussion 

Key results 18 Summarise key results with reference to study objectives 16 278-286

Limitations 19 Discuss limitations of the study, taking into account sources of potential bias or imprecision. Discuss both direction and magnitude of any potential bias 21 394-408

Interpretation 20 Give a cautious overall interpretation of results considering objectives, limitations, multiplicity of analyses, results from similar studies, and other relevant evidence 17-21 287-408

Generalisability 21 Discuss the generalisability (external validity) of the study results 21 404-406

Other information 

Funding 22 Give the source of funding and the role of the funders for the present study and, if applicable, for the original study on which the present article is based We included funding information in the editorial system.

Q-9: Abstract methods: you conducted a ‘multivariable’, not ‘multivariate’, logistic regression analysis.

A-9: Authors’ response: Thank you for your question. We corrected the error as you commented. To reflect the reviewer’s comment,, we revised the previous sentence from Abstract, “Descriptive statistics were prepared and multivariate logistic regression analysis performed.” to read as follows: “Descriptive statistics and multivariable logistic regression analyses were performed.” [Abstract, Line 34-35, Page 2]

Newly added references

22. Conti AA, McLean L, Tolomeo S, Steele JD, Baldacchino A. Chronic tobacco smoking and neuropsychological impairments: a systematic review and meta-analysis. Neurosci Biobehav Rev. 2019; 96:143‐154. pmid:30502351

23. Evert, D.L., and Oscar-Berman, M. Alcohol-related cognitive impairments: An overview of how alcoholism may affect the workings of the brain. Alcohol Health Res World. 19(2):89-96, 1995.

24. Favieri F, Forte G, Casagrande M. The Executive Functions in Overweight and Obesity: a Systematic Review of Neuropsychological Cross-Sectional and Longitudinal Studies. Front Psychol. 2019; 10:2126. pmid: 31616340

25. Dye L, Boyle NB, Champ C, Lawton C. The relationship between obesity and cognitive health and decline. Proc Nutr Soc. 2017; 76(4):443‐454. pmid:28889822

26. Uhlmann RF, Larson EB, Koepsell TD, Rees TS, Duckert LG. Visual impairment and cognitive dysfunction in Alzheimer’s Disease. J Gen Intern Med. 1991;6(2):126-132. https://doi.org/10.1007/bf02598307.

29. Cuevas HE. Type 2 diabetes and cognitive dysfunction in minorities: a review of the literature. Ethn Health. 2019; 24(5):512‐526. pmid:28658961

34. Peters R. Ageing and the brain. Postgrad Med J. 2006; 82(964):84-8. pmid: 16461469

35. Forte G, De Pascalis V, Favieri F, Casagrande M. Effects of Blood Pressure on Cognitive Performance: a Systematic Review. J Clin Med. 2019; 9(1):34. pmid: 31877865

41. Saw YM, Than TM, Win EM, et al. Myanmar language version of the Revised Hasegawa’s Dementia Scale. Nagoya J Med Sci. 2018; 80(4):435-450. pmid:30587859

Additional revision

1. To reflect the reviewers’ comments, we revised the references by adding 9 new references and removing 6 old references. The total number of the references in the revised manuscript becomes 44. [References: Page 23-27]

2. We revised the abstract as follows: “Abstract

Background: Globally, elderly population with impaired cognitive function, such as dementia, has been accelerating, and Myanmar is no exception. However, cognitive function among elderly in Myanmar has rarely been assessed. This study aimed to identify the rate of cognitive impairment and its risk factors among the elderly in Myanmar. 

Methods: This cross-sectional study was conducted at rural health centers in Nay Pyi Taw Union Territory, Myanmar, from December 2018 to January 2019. In total, 757 elderly individuals aged 60 years or over (males: 246 [32.5%], females: 511 [67.5%]) were interviewed using a face-to-face method with a pre-tested questionnaire Descriptive statistics and multivariable logistic regression analyses were performed. 

Results: The rate of impaired cognitive function among participants was 29.9% (males: 23.6%; females: 32.9%). The following participants were more likely to present cognitive impairment: those aged 70–79 years (adjusted odds ratio [AOR] = 1.8; 95% confidence interval [CI]: 1.19–2.70) and 80 years or older (AOR = 3.9; 95% CI: 2.25–6.76); those who were illiterate (AOR = 9.1; 95% CI: 3.82–21.51); and those dependent on family members (AOR = 1.6; 95% CI: 1.04–2.44). The elderly livening with their families and those who reported having good health (AOR = 0.7; 95% CI: 0.44–0.99) were less likely to have cognitive impairment.

Conclusion: Using the HDS-R Myanmar version, this study reported that there out of five elderly participants had cognitive impairment, and its risk factors, altering policy makers that Myanmar needs to prepare for adequate healthcare services and social support for elderly with cognitive impairment. Future research should be performed not only to detect general cognitive impairment but also to differentiate specific cognitive domains impairments among Myanmar elderly. Longitudinal studies are needed to observe the causal and protective factors associated with cognitive impairments in Myanmar.” [Abstract, Line 27-50, Page 2]

3. In the fifth paragraph of the Results section describing for Table 4, three-generation family (UOR = 0.4; 95% CI: 0.17–0.94) was changed into skip-generation family (UOR = 0.4; 95% CI: 0.17–0.94). [Results, Line 257-258, Page 14]

---

## [Decision Letter · Decision Letter 1]

22 Jun 2020

PONE-D-20-03550R1

Cognitive impairment and its risk factors among Myanmar elderly using the Revised Hasegawa’s Dementia Scale: A cross-sectional study in Nay Pyi Taw, Myanmar

PLOS ONE

Dear Dr. Saw,

Thank you for submitting your manuscript to PLOS ONE. After careful consideration, we feel that it has merit but does not fully meet PLOS ONE’s publication criteria as it currently stands. Therefore, we invite you to submit a revised version of the manuscript that addresses all the points raised during the review process.

We look forward to receiving your revised manuscript.

Kind regards,

Gianluigi Forloni

Academic Editor

PLOS ONE

Reviewers' comments:

Reviewer's Responses to Questions

**Comments to the Author**

1. If the authors have adequately addressed your comments raised in a previous round of review and you feel that this manuscript is now acceptable for publication, you may indicate that here to bypass the “Comments to the Author” section, enter your conflict of interest statement in the “Confidential to Editor” section, and submit your "Accept" recommendation.

Reviewer #1: All comments have been addressed

Reviewer #2: (No Response)

2. Is the manuscript technically sound, and do the data support the conclusions?

Reviewer #1: Yes

Reviewer #2: Yes

3. Has the statistical analysis been performed appropriately and rigorously? 

Reviewer #1: Yes

Reviewer #2: Yes

4. Have the authors made all data underlying the findings in their manuscript fully available?

Reviewer #1: (No Response)

Reviewer #2: Yes

5. Is the manuscript presented in an intelligible fashion and written in standard English?

Reviewer #1: (No Response)

Reviewer #2: Yes

6. Review Comments to the Author

Reviewer #1: (No Response)

Reviewer #2: The manuscript is much improved from the first round, though the authors still need to address a few issues.

1. I am still unclear by what the authors mean when they write 'unable to subtract the value'. Please explain, in the manuscript, what you mean by 'value'. Are you referring to the ability to perform simple arithmetic?

2. The authors' response to my question about why they stratified the HDS-R should be added to the manuscript.

3. The chi-square critical values in Table 2 do not communicate useful information. Please report the exact p-values instead.

4. Since the study is cross-sectional, the authors cannot write about 'risk'. In the second paragraph of the discussion, they should replace 'risk' with 'odds'. The same edit should be made in other areas of the manuscript where the term risk may occur.

5. Please explain how missing data were handled in the regression analysis.

6. In the limitations section, the authors write that domain-specific cognitive impairment could not be assessed because a psychiatrist did not assess the participants. However, various cognitive tests exist to permit the assessment of specific cognitive domains, so the authors should say that the HDS-R allowed them to assess global cognitive impairment, and they were unable to measure impairment in specific domains.

7. I am uncertain of the value of figure 1 - suggest deletion.

8. many of the odds ratios in Table 4 are still fairly wide, despite the large sample size. The authors should add mention of this fact to the limitations.

9. At the end of the Ethical considerations section, the authors should mention that they reported their study in conjunction with the STROBE guidelines. Please cite the guidelines and include the STROBE checklist as an appendix.

7. PLOS authors have the option to publish the peer review history of their article (what does this mean?). If published, this will include your full peer review and any attached files.

Reviewer #1: No

Reviewer #2: No

---

## [Author Response · Author response to Decision Letter 1]

7 Jul 2020

Response letter (Response to Reviewers)

PONE-D-20-03550: Cognitive impairment and its risk factors among Myanmar elderly using the Revised Hasegawa’s Dementia Scale: A cross-sectional study in Nay Pyi Taw, Myanmar

Thank you very much editor and reviewers for your valuable comments and suggestions. We have revised the manuscript accordingly. The revised and edited sentences (and words) are mentioned using a track-changes function in the revised manuscript. We also submitted a clean version of revised manuscript as a separate file. In below responses, we noted reviewer’s comments in black color and our responses in blue color. 

Reviewer #1: 

Reviewer #1: All comments have been addressed.

Authors’ response: Thank you very much for your insightful comments and suggestions on our manuscript, which has significantly improved by your constructive comments and suggestions. 

Reviewer #2: 

Reviewer #2: The manuscript is much improved from the first round, though the authors still need to address a few issues.

Authors’ response: Thank you very much for your careful reading of our manuscript and providing valuable comments and suggestions, which have been very helpful in improving our manuscript. We have revised our manuscript according to your comments and suggestions point by point. 

Q-1. I am still unclear by what the authors mean when they write 'unable to subtract the value'. Please explain, in the manuscript, what you mean by 'value'. Are you referring to the ability to perform simple arithmetic?

A-1: Authors’ response: Thank you very much for your comments. Yes, we would like to mention the ability to perform simple arithmetic calculation. The participant needs perform serial subtractions of 7s. The question is "subtract 7 from 100". As suggested, we revised our previous sentences “Elderly individuals who lived in the study area less than six months, who were diagnosed with cognitive impairments along with mental and physical disorders (seriously ill), and who did not understand the Myanmar language and were unable to subtract the value were excluded from the study.” to read as follows: “Elderly individuals who lived in the study area less than six months, those who were diagnosed with cognitive impairments along with mental and physical disorders (seriously ill), those who did not understand the Myanmar language, and those who were unable to perform simple arithmetic calculation were excluded from the study.” [Methods, Line 135-139, Page 6]

Q-2. The authors' response to my question about why they stratified the HDS-R should be added to the manuscript.

A-2: Authors’ response: Thank you very much for your comments. As suggested, we added information of why we stratified the HDS-R in the manuscript as follows: “The dependent variable was cognitive impairment, which was dichotomous into “≤20 points” (presence of cognitive impairment) and “≥21 points” (absence of cognitive impairment). The cut-off point, 20/21, was applied based on the evaluation study of HDS-R test reporting 0.90 for sensitivity and 0.82 for specificity [42]. [Methods, Line 168-171, Page 7-8]

Q-3. The chi-square critical values in Table 2 do not communicate useful information. Please report the exact p-values instead.

A-3: Authors’ response: Thank you very much for your comments. We have reported exact p-values in the Table 1 and Table 2. [Results: Table 1, Page 10-11; Table 2, Page 12]

Q-4. Since the study is cross-sectional, the authors cannot write about 'risk'. In the second paragraph of the discussion, they should replace 'risk' with 'odds'. The same edit should be made in other areas of the manuscript where the term risk may occur.

A-4: Authors’ response: Thank you very much for your valuable comments. We have replaced 'risk' with 'odds' throughout the revised manuscript. [Discussion, Line 279, Page 17; Line 301, Page 18; Line 325, Page 19; Line 347, 358, Page 20; Line 363, Page 21]

Q-5. Please explain how missing data were handled in the regression analysis.

A-5: Authors’ response: Thank you very much for your comments. During data cleaning stage, we excluded 54 participants/cases those who missed to answer the dependent and independent variables. As suggested, we amended previous sentence “After cleaning the data and removing those with missing responses to the main outcome variables, 757 elderly participants (males: 246 [32.5%], females: 511 [67.5%]) were considered for the final data analysis.” to read as follows: “After cleaning the data and removing those with missing responses to the dependent and independent variables, 757 elderly participants (males: 246 [32.5%], females: 511 [67.5%]) were considered for the final data analysis.” [Methods, Line 154-157, Page 7]

Q-6. In the limitations section, the authors write that domain-specific cognitive impairment could not be assessed because a psychiatrist did not assess the participants. However, various cognitive tests exist to permit the assessment of specific cognitive domains, so the authors should say that the HDS-R allowed them to assess global cognitive impairment, and they were unable to measure impairment in specific domains.

A-6: Authors’ response: Thank you very much for your comments. As suggested, we revised the previous sentence from limitation section “However, cognitive impairment among the participants in this study was not confirmed by a psychiatrist; therefore, cognitive impairment in specific domains such as executive function, spatial working memory, processing speed, attention, verbal memory, or verbal fluency could not be ruled out.” to read as follows: “However, cognitive impairment among the participants in this study was not confirmed by a psychiatrist. Therefore, cognitive impairment in specific domains such as executive function, spatial working memory, processing speed, attention, and verbal memory or verbal fluency could not be ruled out regardless of HDS-R allowance to assess cognitive impairment.” [Discussion-limitation, Line 393-397, Page 22]

Q-7. I am uncertain of the value of figure 1 - suggest deletion.

A-7: Authors’ response: Thank you very much for your comments. As suggested, we deleted figure 1. 

Q-8. many of the odds ratios in Table 4 are still fairly wide, despite the large sample size. The authors should add mention of this fact to the limitations.

A-8: Authors’ response: Thank you very much for your valuable comments. As suggested, we have added below sentence to limitation “Furthermore, the fairly wide confidence intervals were observed for some predictors despite the large sample size.” [Discussion-limitation, Line 397-398, Page 22] 

Q-9. At the end of the Ethical considerations section, the authors should mention that they reported their study in conjunction with the STROBE guidelines. Please cite the guidelines and include the STROBE checklist as an appendix.

A-9: Authors’ response: Thank you very much for your valuable suggestion. As suggested, at the end of the Ethical consideration section, we added STROBE checklist information as follows: “Furthermore, this study followed the Strengthening the Reporting of Observational Studies in Epidemiology (STROBE) guidelines (Appendix 1).” [Methods, Line 212-213, Page 9]

Newly added reference

1. Imai Y, Hasegawa K. The revised Hasegawa’s Dementia Scale (HDS-R)- Evaluation of its usefulness as a screening test for dementia. J Hong Kong Coll. Psychiatr .1994; 4:20-24. 

Additional revision

1. We have corrected HSR-D to HDS-R in introduction section. [Introduction, Line 116-117, Page 5]

2. We amended the journal title abbreviations of reference no. 10,12,13, and 16.

---

## [Editor Report · Decision Letter 2]

13 Jul 2020

Cognitive impairment and its risk factors among Myanmar elderly using the Revised Hasegawa’s Dementia Scale: A cross-sectional study in Nay Pyi Taw, Myanmar

PONE-D-20-03550R2

Dear Dr. Saw,

We’re pleased to inform you that your manuscript has been judged scientifically suitable for publication and will be formally accepted for publication once it meets all outstanding technical requirements.

Kind regards,

Gianluigi Forloni

Academic Editor

PLOS ONE
---

## [Editor Report · Acceptance letter]

16 Jul 2020

PONE-D-20-03550R2 

Cognitive impairment and its risk factors among Myanmar elderly using the Revised Hasegawa’s Dementia Scale: A cross-sectional study in Nay Pyi Taw, Myanmar 

Dear Dr. Saw:

I'm pleased to inform you that your manuscript has been deemed suitable for publication in PLOS ONE. Congratulations! Your manuscript is now with our production department. 

Kind regards, 

on behalf of

Dr. Gianluigi Forloni 

Academic Editor

PLOS ONE